# Uncertainty quantification in clinical settings: A retinal fundus screening study and benchmarking

## Abstract

We offer the most extensive benchmark for uncertainty quantification (UQ) in retinal AI screening, providing practical guidance for clinical evaluators/regulators and highlighting the importance of risk–coverage–accuracy analysis. We methodically assess six well-known post-hoc UQ techniques in three main diseases: glaucoma (115K+ images), age-related macular degeneration (29K+ images), and diabetic retinopathy (105K+ images). Our benchmark comprises three Vision Transformer variations, standardized train/test/calibration splits, and evaluation on both public datasets and in-house clinical data from a local hospital. Results show that screening models can be miscalibrated and overconfident, and although UQ is helpful, its benefits are highly method- and disease-dependent. Our risk–coverage–accuracy analysis shows coverage drastically decreases as risk limits increase, and no single approach is consistently dependable in all contexts. While neither method consistently outperforms the others, Deep Ensembles and Test-Time Augmentation (TTA) are the two practical UQ approaches that most frequently enhance selective prediction and/or calibration. Conformal Prediction (CP) serves as a must-have safety rail, ensuring alignment between nominal and observed coverage. However, no method can reliably achieve the 2% target-risk required for autonomous screening without sacrificing coverage. These findings highlight the need for more robust post-hoc UQ methods, both for in-distribution scenarios and under domain shifts (out-of-distribution), as well as improved mechanisms for capturing disagreements and implementing policy-aware thresholding in human-in-the-loop workflows. To facilitate progress in this field, we release our benchmark, which includes standardized data splits, trained model checkpoints, code, and an online demo for interactive exploration, thereby providing a reference for future UQ research in ophthalmic AI screening.

## 1 Introduction

The application of Artificial Intelligence (AI) in healthcare holds significant clinical importance, particularly for tasks like early disease screening and automating medical image analysis. However, despite this potential, the reliability and trustworthiness of these sophisticated systems remain critical concerns, currently limiting their widespread deployment in real clinical scenarios where patient safety is paramount Khan et al. (2025); Kim et al. (2023); Rajpurkar et al. (2022). A primary reason for these concerns is the inherent uncertainty associated with AI predictions. AI systems operate using complex models and large datasets, and factors such as inherent noise within the data and the limitations of the models themselves lead to unavoidable uncertainty Wang et al. (2025).

As we delve deeper into the nuances of AI-based medical image screening, it becomes increasingly clear that uncertainty quantification (UQ) is not merely an ancillary consideration but a fundamental necessity for effective deployment in real-world clinical settings. For effective machine-assisted medical decision-making, quantifying uncertainty per patient/case is vital. When faced with ambiguity, AI should be capable of abstaining from predictions,

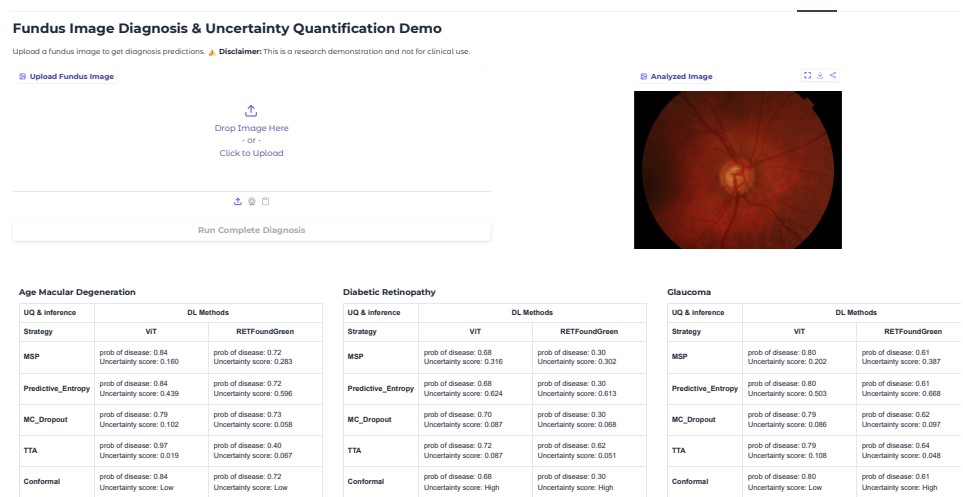

Figure 1: A snapshot of the deployed online demo with trained models and UQ methods, freely accessible on Hugging Face Spaces (`http://`). Notice that in this example, even though the fovea is not fully visible, the AMD model still makes an overconfident prediction. We can also see inconsistencies between the outputs of different models trained for the same disease but with different random weight initializations. Uncertainty scores can help improve the final decision in some inference cases, but not always, and in some cases even incorrect predictions are assigned a low uncertainty score. Deep ensembles were omitted due to limited online storage capacity.

seeking human expertise (i.e., "learning to defer"), or collecting additional data Zou et al. (2023); Kompa et al. (2021); Begoli et al. (2019); Challen et al. (2019); Alves et al. (2025).

Deep learning models have great potential for automating medical image analysis, but ignoring uncertainty can pose serious risks in clinical decision-making, undermining trust in these systems. Therefore, it is crucial to rigorously evaluate and benchmark UQ methods to enhance the reliability of deep learning models in medical imaging.

It is important to clarify that our focus is not on creating uncertainty-aware deep learning models solely to improve their performance. Instead, we are interested in understanding how to effectively utilize developed models and transfer them to clinical settings during the inference phase, evaluating them on a per-sample, per-image, and per-patient basis (**post-hoc**), and providing guidance for researchers and evaluators on the clinical side. To save space, the related work section has been moved to the appendix. In summary, while there are a few uncertainty-aware studies aimed at improving the performance of trained models for fundus-based diagnosis, they fall outside the scope of our investigation, and their datasets are limited with no benchmarking.

**Objectives and contributions:** By providing this evidence-based comparison, we aim to equip clinical researchers, regulators, and evaluators with the knowledge needed to assess the reliability and trustworthiness of AI models in clinical settings, particularly regarding risk–coverage–accuracy trade-offs, which also benefits AI developers. While this work focuses on retinal image–based screening applications, the shared code and insights can be valuable for other domains as well. The primary objectives and contributions of this benchmark are: **1)** Conduct a systematic evaluation of six post-hoc UQ methods across three major retinal diseases—age-related macular degeneration (AMD), glaucoma, and diabetic retinopathy (DR). **2)** Establish a large-scale, multi-disease benchmark with standardized train/test/calibration splits, ensuring reproducibility and fair comparison. **3)** Quantify the gap between laboratory conditions and real-world deployment (lab to clinic) by testing methods on both public datasets (in-distribution) and a local clinical dataset with physician annotations (out-of-distribution). **4)** Analyze whether clinically viable screening ($<2\%$ target risk) is achievable, highlighting trade-offs between safety, coverage, and practical utility. **5)** Assess calibration and statistical validity of UQ methods, including reliability diagrams and conformal prediction coverage. **6)** Investigate alignment between UQ outputs and clinical complexity by testing their ability to detect cases with physician disagreement. **7)** Release

Table 1: Comparison of uncertainty quantification methods. Note: The input for the conformal prediction/inference method consists of prediction values, allowing it to be applied on top of each prediction generated by other methods.

| | None | MSP / Entropy | TTA | MC Dropout | Conformal | Deep Ensemble | | | |
| --- | --- | --- | --- | --- | --- | --- | --- | --- | --- |
| | | | | | | Var | Aleatoric | Epistemic | Total |
| Prediction value | $\hat{y}$ | $\hat{y}$ | Avg. over augmentations | Avg. over dropout passes | Set prediction with coverage | Avg. over models | same | same | same |
| Uncertainty | – | $1 - \max(p_i)$ or $-\sum p_i \log p_i$ | | | Length of prediction set | | $\mathbb{E}[\mathrm{Var}[p|x]]$ | $\mathrm{Var}[\mathbb{E}[p|x]]$ | Sum of aleatoric and epistemic |
| Uncertainty (Classwise: +) | – | | Variance across aug. | Variance across dropout | | Variance across models | | | |
| Inference strategy | Single pass | Single pass | Multiple augmentations | Multiple stochastic passes | Calibration set + test prediction | Multiple models | same | same | same |

open-sourced trained screening models with integrated UQ, online demos, and evaluation code as open-source resources to accelerate research.

## 2 METHODOLOGY AND EXPERIMENTAL DETAILS

Dataset: To achieve our objectives, we compiled a large collection of globally available fundus photo datasets that include labels or can have labels extracted for glaucoma, diabetic retinopathy, and age-related macular degeneration. This effort resulted in the accumulation of over 100,000 images for diabetic retinopathy, 28,000 for age-related macular degeneration, and 114,000 for glaucoma, as reported in Table 3(appendix).

From these datasets, only the samples designated as test sets by the original publishers were used for testing; otherwise, the photos were utilized for training. In addition, since the conformal prediction method requires a calibration set, we randomly selected a subset of photos from the test sets to serve as the calibration set (20%), ensuring that the selection was stratified. This calibration set will be excluded from all evaluations and metrics. The CSV files related to the train, test, and calibration sets are available as a benchmark in our GitHub repository to enhance reproducibility. All labels are provided as binary values for screening (0 for healthy, 1 for referral). To our knowledge, this is the largest benchmark training and evaluation set available, encompassing datasets from around the world.

For further investigation, we also utilized a local dataset from Hospital X that has been annotated for glaucoma presence and referral by three ophthalmologists. This dataset contains 536 photos of acceptable quality and will be used as an external and out-of-distribution test set. Similarly, 20% of this dataset has been designated as the calibration set.

Deep Learning Models: We selected three Vision Transformer (ViT) variations as the backbones for our experiments, all from the timm library and leveraging powerful pretrained weights Wightman (2019): ViT: A benchmark Vision Transformer model based on the DINOv2 self-supervised learning paradigm, pretrained on the ImageNet dataset Oquab et al. (2024). RETFound-Green: This model employs the same architecture but is a foundation model specifically pretrained on a massive dataset of 1.6 million unlabeled fundus images, making it particularly suitable for retinal tasks. We used the publicly available weights for initialization Engelmann and Bernabeu (2025). ViT(FLSD-53): Mukhoti et al. showed that using focal loss instead of cross-entropy can improve model calibration Mukhoti et al. (2020). They also demonstrated that the sample-dependent variant *FLSD-53* where the hardest and most uncertain examples ($\hat{p}_y \in [0, 0.2)$) receive a stronger focus ($\gamma = 5$), and less difficult samples ($\hat{p}_y \in [0.2, 1]$) receive a lower focus ($\gamma = 3$) outperforms even temperature-scaled models. Based on these findings, we selected this approach to train a ViT model as an uncertainty-aware and calibration-oriented baseline. For all models, we employ a transfer learning strategy in which the pretrained feature extraction layers are frozen. Only a new custom classification head, consisting of a single linear layer that maps the feature dimension to the number of output classes (2 for binary classification: healthy vs. referral), is trained. The dropout rate in this layer is set at 10%. The input image size for the deep learning models was set to 392 by 392 pixels.

Image Reprocessing: All fundus photos undergo a multi-step preprocessing pipeline before being passed to the models. This preprocessing ensures that the retina is present within the photo, extracts the region of interest, and ultimately resizes the photo to a square format

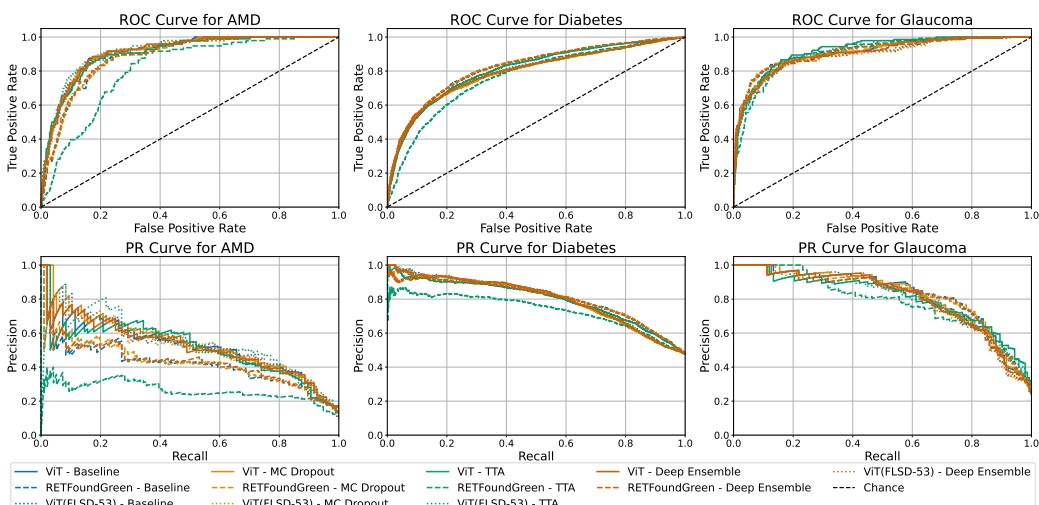

Figure 2: Figures of ROC and PR curves for different trained models and inference strategies.

(equal height and width) using zero padding. The scripts used for this process are publicly available EyeQ Fu et al. (2019) (code at Fu (2019)). Also, all photos, for both training and testing, are normalized using the standard ImageNet mean and standard deviation. Augmentations: During training, we improve robustness through data augmentations: images are resized to 110% and randomly cropped to target resolution, with random flips, small rotations ($\leq 10$), color jitter (brightness, contrast, saturation, hue), and Gaussian blur. For validation and testing, we resize images to the final input resolution without random augmentations, but we apply the same training augmentations for uncertainty quantification at test time. Training Protocol: is available at appendix.

Uncertainty Quantification Methods: We implement and compare six distinct uncertainty quantification (UQ) methods, as shown in Table 1. For Monte Carlo Dropout, we perform (T = 50) stochastic forward passes for each input image. For the deep ensemble method, (N = 5) independent models are trained with dropout enabled, and the train/validation set is randomly initialized for each model's training. For test time augmentation (TTA), each test photo is accompanied by (K = 20) augmented photos that are fed to the model, following the augmentation strategy described above. The justification for these parameters is provided in the appendix.

Evaluation Metrics: We use a comprehensive set of metrics to evaluate predictive performance and uncertainty quality (extended description is available in the appendix). As classification metrics, AUROC($\uparrow$) and AUPRC($\uparrow$) measure model discriminative ability, while PPV($\uparrow$) and NPV($\uparrow$) assess predictive values. In the context of calibration metrics, ECE($\downarrow$) quantifies alignment between predicted confidence and actual accuracy, NLL($\downarrow$) evaluates probabilistic prediction quality, and the Brier Score($\downarrow$) assesses calibration and sharpness. For uncertainty metrics, AURC($\downarrow$) measures the effectiveness of uncertainty estimates, Risk@90%($\downarrow$) Coverage reports error rates for low-uncertainty samples, and Coverage@5% Risk($\uparrow$) indicates the fraction of samples processed automatically while maintaining low error rates. Furthermore, visually, ROC and precision-recall curves illustrate model performance, reliability diagrams assess calibration quality, risk-coverage curves show the relationship between coverage and risk, and conformal prediction coverage plots evaluate statistical validity.

## 3 RESULTS

### 3.1 DETECTION AND CALIBRATION PERFORMANCE

First, we establish baseline classification performance on the public test set, as shown in Figure 2 and Table 4. Our results indicate that most methods achieved high AUROC

scores and acceptable AUPRC values. The glaucoma and diabetic retinopathy models show promising screening performance but are not ready for deployment. In contrast, the low AUPRC for AMD indicates unreliable identification of positive cases, despite a good AUROC score, and improving this is beyond the scope of this study. For UQ-enabled methods, averaged predictions are used. TTA consistently improves performance, especially with ViTs on AMD and glaucoma, while MC Dropout and Deep Ensembles add little. RETFound-Green performs well for diabetes but is highly TTA-sensitive, leading to severe drops on AMD, suggesting less robust features.

Our calibration investigation, reported in Figure 3(reliability diagrams) and Table 4, shows that overall calibration improvements are disease- and model-dependent. However, TTA stands out as the most effective strategy for enhancing model confidence alignment in AMD and glaucoma. Additionally, our models for predicting AMD exhibit the worst calibration and demonstrate overconfidence. Furthermore, surprisingly, the ViT (FLSD–53) model—which was expected to produce more calibrated outcomes—resulted in worse calibration in our case study. This may be due to several hyperparameters (e.g., threshold of $\hat{p}_y$ such as 0.2, or values of $\gamma \in \{3, 5\}$), while the original experiments were conducted on substantially different datasets, including CIFAR-10 (10 classes, 60,000 images), CIFAR-100 (100 classes, 60,000 images), Tiny-ImageNet (200 classes, 110,000 images), and ImageNet (ILSVRC-2012: 1,000 classes, about 1.2M images). These findings again highlight the importance of calibration and hyperparameter tuning, particularly when adapting models under domain shift. Optimizing these parameters was outside the scope of this study.

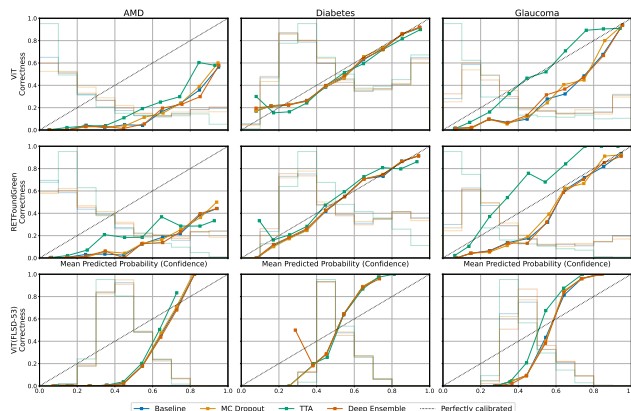

Figure 3: Reliability diagrams for disease detection models, where well-calibrated methods lie near the diagonal.

## 3.2 Uncertainty Analysis

Figure 4 and Table 5 show the information obtained from risk-coverage (selective prediction) analysis to assess the effectiveness of UQ methods and their impacts. Conformal prediction will be discussed in the next subsection due to its distinct nature. The variations in the distribution of uncertainty scores for each UQ method are shown in Figure 9.

Glaucoma benefits the most from uncertainty-based selective prediction (for both ViT and RETFound-Green), with a very low AURC (baseline 0.191, dropping to 0.046 with deep ensemble) and the lowest Risk@90% Coverage ( 0.101). In contrast, Coverage@5% Risk is high (>0.6), indicating that uncertainty signals effectively identify unreliable cases. For glaucoma, uncertainty-based methods significantly enhance both efficiency and safety, demonstrating a clear benefit from selective prediction. In AMD, there are large improvements in AURC and Risk@90% Coverage, but unstable coverage at low-risk thresholds persists. The amount of improvement is much greater with the RETFound model (AURC improves from 0.457 to 0.082). Variance ensembles are the only method achieving both better risk and meaningful coverage, indicating a dramatic gain, though likely at the cost of coverage. In diabetes, there are modest improvements in AURC and Risk@90% Coverage (for both ViT and RETFound), but coverage remains generally poor, suggesting that uncertainty signals are less informative. Overall, improvements are modest and not robust.

Variance-based deep ensembles emerge as the most effective UQ method, providing clear improvements for AMD and glaucoma across all metrics. Aleatoric and epistemic ensembles help in diabetes and glaucoma, but with less consistency. In contrast, Maximum Softmax Probability (MSP), entropy, dropout, and test-time augmentation (TTA) offer only partial gains and often compromise coverage.

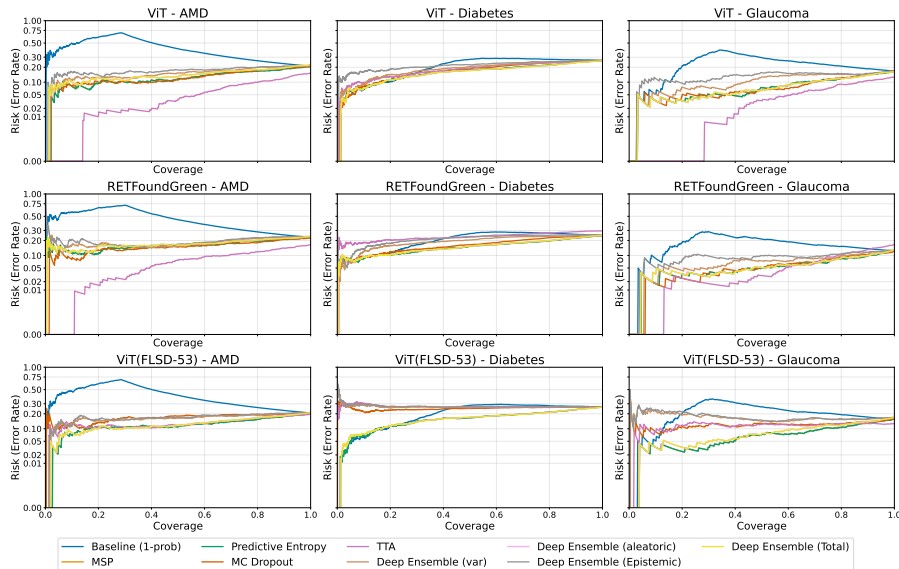

Figure 4: Figures of Risk–Coverage analysis showing the effectiveness of UQ methods. The x-axis shows coverage, the proportion of test instances the system predicts automatically, with the rest rejected (abstained/referred) for expert review. For example, at 90% coverage, the 10% of test samples with the highest uncertainty scores are removed and referred to an expert.

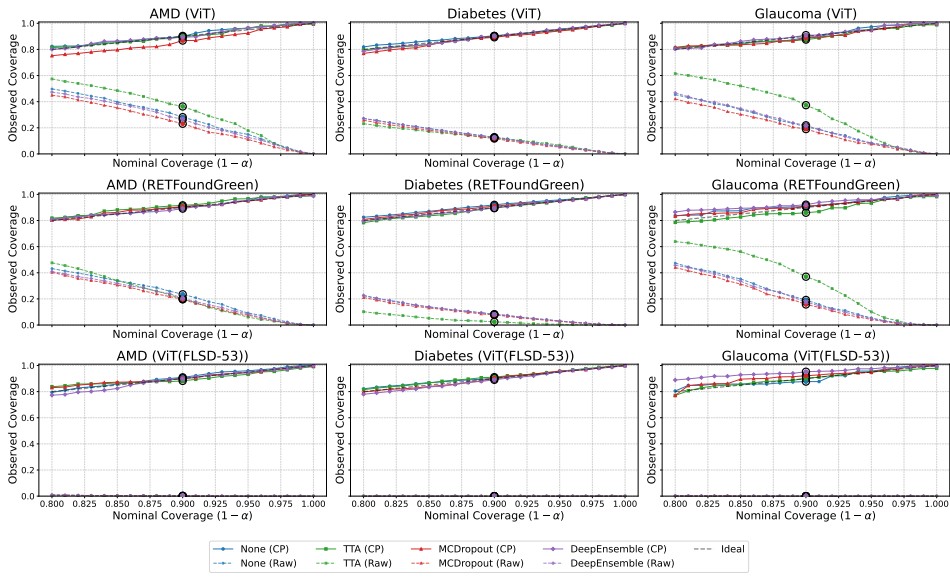

Figure 5: Observed vs. nominal coverage showing under-coverage in raw predictions and correction with CP. All the raw methods exhibit undercoverage and overconfidence, whereas CP effectively compensates for these shortcomings.

## 3.3 CONFORMAL PREDICTION/INFERENCE

As mentioned previously, conformal prediction (CP) inference can be applied on top of all other methods, using prediction values as input. Figure 5 presents the Conformal Prediction Coverage Plot (Observed vs. Nominal Coverage). The Observed Coverage vs. Nominal Coverage curve shows how often the conformal prediction sets actually contain the true label compared to the reliability level is requested. If observed coverage stays at or above the nominal values, the method is reliable or conservative; if observed coverage drops below the nominal values, the method is overconfident and does not meet the desired guarantee.

As shown in Figure 5, all raw models (without CP) display a significant gap between nominal and observed coverage and as expected, the worst-performing model is ViT (FLSD–53), which exhibited the poorest calibration performance. Consequently, its coverage beyond 80% is nearly zero; however, CP substantially improves its performance. As an example, for the ViT-Glacuoma, when the raw model claims 90% coverage, it actually only covers about 20–40% of true outcomes, reflecting severe overconfidence and leading to undercoverage in real setting. CP addresses this issue by pulling the observed coverage curves closer to the ideal diagonal. With CP, the gap largely disappears, and coverage approaches nominal levels across diseases and methods.

The reason for this gap is that modern neural classifiers are often miscalibrated, particularly in medical imaging, where raw softmax probabilities are not reliable confidence estimates. The substantial gap between raw and CP curves illustrates **how unreliable uncalibrated model confidences are, and why conformal prediction is crucial**: it transforms overconfident raw probabilities into valid guarantees, ensuring that observed coverage aligns with the desired nominal level. Table 6 (showing only 90% coverage, i.e., $\alpha = 0.1$) illustrates the impact of conformal prediction on calibration. We found that CP consistently enforces near-nominal coverage ( 0.90) across models, methods, and diseases, effectively addressing the severe undercoverage observed in raw predictions.

### 3.4 Performance Under Clinical Domain Shift

For further investigation, we evaluated the performance of the models and UQ methods using a local dataset containing 743 test photos related to glaucoma. Figure 6 presents the curves and figures of detection performance (ROC, PR), the calibration reliability diagram, and the achieved risk-coverage (Detailed tables can be found in Appendix Table 7.).

Comparing our local hospital glaucoma results (out-of-distribution) with the earlier test-set (i.e. publicly available datasets, which are in-distribution) performance: Previously, AUROC for glaucoma was very strong (0.905–0.928) and AUPRC was similarly high (0.80–0.818). On the local hospital data, both metrics drop notably: AUROC falls to 0.77 (ViT) and 0.64 (RETFoundGreen), while AUPRC decreases to  0.74 (ViT) and 0.66 (RETFoundGreen). On local hospital glaucoma data, detection performance drops sharply, particularly for RETFound-Green, indicating strong domain shift, while ViT remains more robust.

As shown, this degradation under domain shift is significant and important. It may be because the training data do not fully represent (not limited to) the hospital's imaging devices (e.g., Topcon, Zeiss), variations in sensor resolution, dynamic range, illumination, flash intensity, and color calibration, as well as differences in patient demographics (e.g., age, race), disease presentations (e.g., under- or overrepresentation of mild cases), or clinical workflows (e.g., with or without dilation). These factors introduce significant domain shifts that reduce model generalization. The amount of degradation may be reduced by considering domain-shift adaptation techniques Zhou et al. (2022), such as test-time adaptation methods (e.g., TENT Wang et al. (2020a) or TTT Sun et al. (2020)), where the model dynamically updates its normalization statistics or minimizes prediction entropy on incoming test streams.

Calibration on local glaucoma data is markedly degraded compared to external test sets, reflecting domain shift. TTA provides the most noticeable correction, especially for ECE in RETFoundGreen, but overall reliability remains weaker than in the original experiments. On local glaucoma data, selective prediction provides only minor improvements over baseline and fails to achieve the strong error–coverage gains observed on in-distribution test sets (AURC is higher (0.27–0.38 vs.  0.05 before), indicating less efficient risk–coverage trade-offs). This indicates a pronounced domain shift, where uncertainty estimates no longer reliably separate correct from incorrect predictions.

On the local glaucoma dataset, both discriminative performance and calibration degrade compared to in-distribution test sets, and uncertainty-based selective prediction provides only a marginal benefit. This **highlights a clear domain shift degradation**, where models remain overconfident and uncertainty estimates lose reliability, limiting their practical utility without further adaptation.

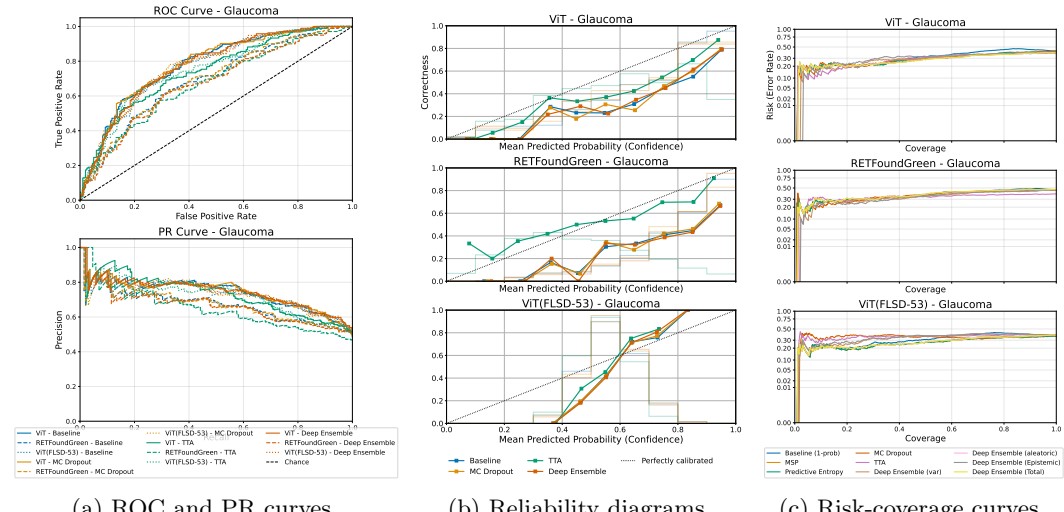

(a) ROC and PR curves.   (b) Reliability diagrams.   (c) Risk-coverage curves.

Figure 6: Achieved results using the trained model on large publicly available datasets applied to the local dataset (glaucoma only). Compared to Figures 2, 3, and 4, there is a clear degradation across all performance metrics, including AUROC, AUPRC, the reliability diagram, and the risk–coverage curve.

### 3.5 Disagreement analysis

One important source of data inconsistency is the disagreement among physicians when establishing ground truth. This represents a serious form of aleatoric uncertainty, typically arising in borderline and clinically challenging cases. To further explore this, we investigated which UQ methods are most effective at detecting such challenging samples and whether their uncertainty aligns with physician disagreement. Only two datasets in our study provide multi-rater labels (both glaucoma): the Drishti dataset, which includes five raters Sivaswamy et al. (2014), and our local dataset, which includes three raters. In both cases, a sample was tagged as "disagreement: yes" if not all ophthalmologists provided the same diagnosis.

Figure 7 displays the violin plots of uncertainty scores for the "disagreement" and "no disagreement" groups, along with the conformal prediction set size for each group (with 2 indicating uncertainty). Monte Carlo Dropout and deep ensemble methods (both aleatoric and total) show a t-test p-value of less than 0.01 for the local dataset (600 test samples after removing the calibration set, with 256 disagreements), indicating a significant difference in uncertainty scores between these groups and suggesting the potential to identify disagreements. However, this finding is not replicated in the Drishti dataset (40 test samples after removing the calibration set, with 19 disagreements). Conformal prediction was unable to detect disagreements. Figure 7 only shows results for the ViT model, while the RETFound-Green and ViT(FLSD-53) models does and does not demonstrate this capability for identifying disagreements, respectively (Figures 10, Figures 11).

## 4 Discussion

Our large-scale benchmark provides several important insights into the role of uncertainty quantification (UQ) in retinal AI screening. 1) Detection performance insights: While classification models achieved strong AUROC values across public datasets, especially for glaucoma and diabetic retinopathy, their calibration and robustness varied substantially across diseases and UQ methods (due to different inference strategy: Table 1). 2) Out-of-distribution performance: Our results reveal a significant gap between laboratory performance and clinical viability for AI-based retinal screening systems. While AUROC scores appear promising (>0.90 glaucoma) over the in-distribution test set, the substantial performance degradation on our local clinical dataset (AUROC dropping to 0.69-0.77 for glaucoma) underscores the persistent challenge of domain shift in medical AI.

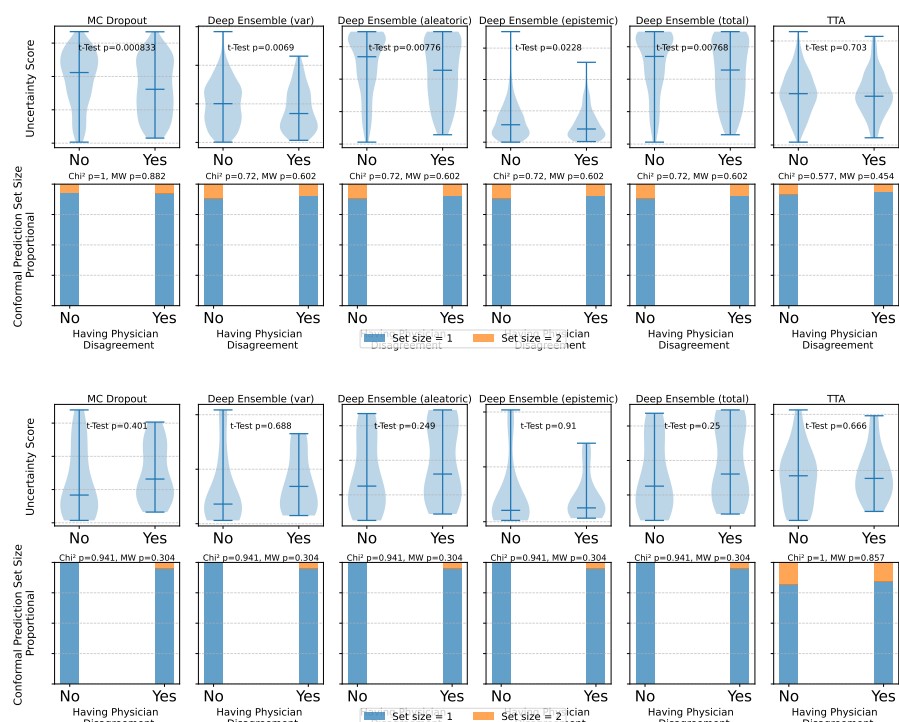

Figure 7: Each subplot shows uncertainty quantification results across disagreement groups (ViT model): top for the local dataset, bottom for Drishti. MC Dropout and deep ensembles show clear separability in uncertainty scores, indicating their ability to detect disagreements.

3) UQ performance: Deep Ensembles emerge as the most reliable approach, providing both improved risk-coverage trade-offs and meaningful uncertainty decomposition into aleatoric and epistemic components. Test-Time Augmentation shows promise for calibration improvement but exhibits model-dependent behavior. Variance-based Deep Ensembles emerged as the most effective method in selective prediction, significantly lowering AURC and risk at fixed coverage, particularly for glaucoma. This suggests that ensemble diversity is a key driver for uncertainty quality in ophthalmic AI. 4) Conformal Prediction as a Critical Safety Net: Our conformal prediction analysis reveals a sobering reality about model overconfidence. The substantial gap between nominal and observed coverage in raw predictions (models claiming 90% confidence while achieving only 10-30% actual coverage) demonstrates dangerous miscalibration that could lead to clinical harm. Conformal prediction's ability to restore statistical validity represents a crucial safety mechanism. 5) Out-of-distribution UQ performance: The performance degradation on our local clinical dataset extends beyond accuracy and calibration breakdown. This suggests that uncertainty estimates themselves become unreliable under domain shift, limiting their protective value. 6) Physician Disagreement Detection: The limited ability of UQ methods to identify cases with physician disagreement (significant only for MC Dropout and deep ensembles on our local dataset) indicates that technical uncertainty measures may not fully capture clinical complexity and ambiguity. 7) Selective Prediction Trade-offs: While uncertainty-based selective prediction improves risk-coverage curves, the coverage rates at clinically relevant risk thresholds remain problematic.

8) The 2% Risk Target: Target risk is the maximum tolerable error for auto-accepted cases; in ophthalmic screening, it is ideally 2% to ensure high sensitivity. Figure 8 presents the target risk analysis (ViT only), with the red dashed line at 0.02 indicating our 2% target risk threshold for clinical deployment. This figure illustrates the performance of the ViT model alongside different UQ methods while sweeping the acceptable risk targets (from p3 to p17, with p1 as the baseline and p2 reflecting the removal of uncertain samples using conformal prediction with ($\alpha = 0.1$)). For each risk target, a threshold (Tau: ($\tau$)) for the uncertainty score is established to meet that target risk (using the calibration set).

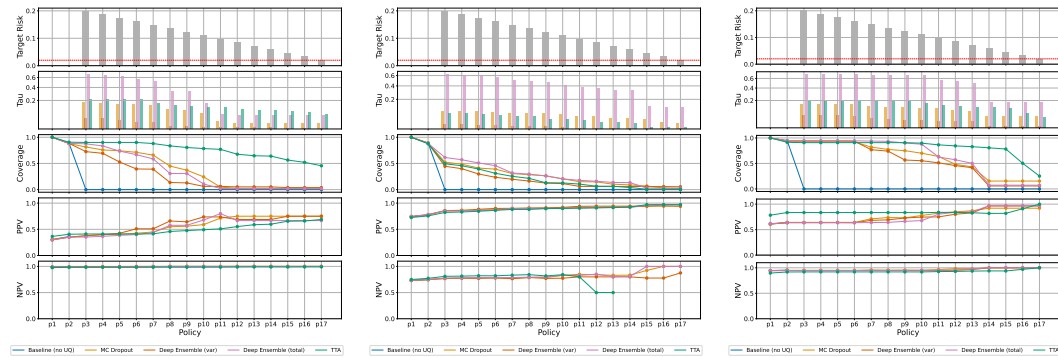

Figure 8: Clinical task target analysis compares ViT with several UQ methods across a sweep of acceptable risk targets (p3–p17; p1 = baseline, p2 = conformal filtering with $\alpha = 0.1$). For each target, a threshold $\tau$ on the uncertainty score is chosen to meet that risk, and the resulting coverage (auto-accepted fraction), PPV, and NPV are reported. Left) AMD, Middle) Diabetics, Right) Glaucoma. larger figures are available in the appendix.

Consequently, the achieved coverage (auto-accepted fraction: ratio of test samples satisfying the policy), PPV, and NPV values are shown. This figure highlights the trade-off between risk targets, acceptable uncertainty scores, coverage, and detection performance. As the risk target decreases, coverage drops significantly, which is problematic for screening utility, while Tau threshold values increase, indicating that higher uncertainty thresholds are necessary. Performance varies among methods, with the baseline (blue) exhibiting the most aggressive drop in coverage, reaching near-zero coverage by policies p3-p4. Deep Ensemble methods (orange/pink) maintain coverage for a longer duration while still reducing risk, whereas TTA (green) demonstrates the most robust coverage retention across policies. Across all three conditions, no method consistently meets the target risk until very restrictive policies (p15-p17) are implemented, and even then, success is sporadic. This suggests that our trained models and UQ methods (while beneficial) are insufficient for fully automated clinical deployment. 9) Limitations of this study: While the advantages of UQ were found to be model-dependent, we only evaluated two deep learning models with frozen feature extractors. Additionally, the sizes of the test and calibration sets are limited, which is common in medical AI research. We applied standard conformal prediction, though adaptive or hierarchical variants may perform better. The training dataset was heavily imbalanced toward the negative class, which the loss function attempted to address. Disagreement analysis was limited to glaucoma and two datasets.

## 5 CONCLUSION

Our large-scale benchmark with detailed risk–coverage–accuracy analysis shows that while uncertainty estimation helps, we are still far from an "automate-and-forget" clinical workflow. No single UQ method is consistently reliable across diseases and models; even at modest risk targets, coverage often collapses, underscoring the gap to clinically realistic, less than 2% risk operation. Conformal Prediction is non-negotiable as a safety measure, as it reliably restores the alignment between nominal and observed coverage that raw models often fail to achieve. Among practical tools, Deep Ensembles offer the most significant gains in selective prediction and help identify challenging cases, while Test-Time Augmentation consistently enhances calibration. However, neither method is uniformly dominant across all scenarios. Real-world deployment continues to be challenging: in the clinical local dataset, both discrimination and calibration decline, while the advantages of uncertainty quantification diminish, highlighting a significant domain shift. We therefore recommend CP as a mandatory layer, with ensembles or TTA on top, and a human-in-the-loop thresholding policy; however, reaching safe, scalable screening will require innovative UQ methods that retain validity under shift, better capture clinician disagreement, and meet strict target-risk constraints. We benchmark and release standardized training/calibration and test splits, share all trained model checkpoints to enable replication, and provide an online demo for interactive exploration.

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

# A  Technical Appendices and Supplementary Material

LLM Usage: Large language models (LLMs) were used only for proofreading, grammar correction, and minor script development. No LLMs contributed to research ideation, experimental design, or substantive writing.

## A.1  Related Work

Our research is situated at the intersection of uncertainty quantification (UQ) in deep learning, its application in medical screening, and its clinical relevance and performance in ophthalmology.

The necessity of responsible AI in medicine has become increasingly clear as healthcare systems integrate artificial intelligence into clinical practice. While AI offers the potential for enhanced diagnostic accuracy and efficiency, it is imperative to recognize the challenges associated with its deployment. Issues such as bias in training data, domain shift between training and real-world settings, and generalization failures can lead to disparities in care and compromised patient safety. Responsible AI practices aim to mitigate these risks by ensuring that AI systems are transparent, fair, and reliable, ultimately fostering trust among healthcare providers and patients alike while supporting equitable access to high-quality medical care Chen et al. (2023); Park et al. (2021); Stetson et al. (2025).

To build upon these principles of responsible AI, it is crucial to employ structured frameworks designed to stress-test and ensure model reliability in real-world clinical applications Antao et al. (2025). Google's Plex framework, for example, provides comprehensive guidelines centered on three core requirements for trustworthy machine learning systems: uncertainty, robust generalization, and adaptation Tran et al. (2022). The first pillar, uncertainty, addresses a model's ability to "know what it doesn't know" Goetz et al. (2024). This is essential for identifying when a model's prediction should be trusted, enabling graceful failures when it is likely to be wrong, and flagging difficult cases for human intervention Goetz et al. (2024). The second pillar, robust generalization, confronts the challenge of distribution shifts, ensuring that a model maintains its performance and reliability when encountering new data from different sources or environments (a common problem when moving from lab to clinic Goetz et al. (2024)). Finally, adaptation evaluates a model's capacity to learn efficiently from new data, a critical feature for systems that must evolve with new clinical information or changing patient populations. By systematically stress-testing models across these three pillars, the Plex framework aims to produce AI systems that are not only accurate but also consistently dependable and safe for deployment in high-stakes medical settings Goetz et al. (2024).

The need for reliable AI has made UQ an active area of research in medical imaging Zou et al. (2023). It has been applied to tasks such as disease classification in radiology Park et al. (2021), tumor segmentation in histopathology Dolezal et al. (2022), and lesion detection in dermatology Yu et al. (2025). These studies consistently show that leveraging uncertainty can identify difficult cases, detect out-of-distribution samples, and enable a "human-in-the-loop" workflow where uncertain predictions are flagged for expert review. However, much of the existing work focuses on a single disease or a single imaging modality. To our knowledge, a large-scale, systematic benchmark of modern UQ methods across three major retinal diseases, particularly with a rigorous evaluation on a real-world clinical dataset, remains a significant gap in the literature. The sources of uncertainty are diverse, ranging from variability in image acquisition (e.g., scanner models, protocols), inherent biological variability between patients, and annotation ambiguity from expert disagreements Alizadehsani et al. (2021); Loftus et al. (2022).

Currently, there are several AI-based screening methods and devices approved by the FDA for diabetic retinopathy (DR), including Digital Diagnostics' IDx DR, EyeNuk's EyeArt, AEYE Health, and iPredict Eye Screening for age-related macular degeneration (AMD). Additionally, Verily Life Sciences LLC, a subsidiary of Alphabet, has announced its CE Mark for DR in India. It is likely that more systems will emerge in the near future.

Uncertainty quantification (UQ) is essential in ophthalmic AI systems to safeguard clinical decision-making. Without calibrated confidence estimates, AI models can become overconfident and mislead clinicians, leading to missed diagnoses or inappropriate referrals Wang et al. (2023); Akram et al. (2025). Past studies underscore this risk: even highly accurate algorithms can struggle when faced with real-world shifts in data. For instance, there is a notable absence of deep learning models that can reliably predict visual fields in clinical settings Eslami et al. (2023). Additionally, the generalization of image analysis across different image-capturing environments Kalahasty et al. (2023) and the transferability of trained models between hospitals and institutions present significant challenges Chuter et al. (2024); Ktena et al. (2024). UQ offers a remedy by flagging low-confidence predictions so that ambiguities or out-of-distribution cases are identified before harm occurs.

In response, a variety of UQ approaches, ranging from Bayesian approximations to conformal prediction, have been developed to create more reliable ophthalmology AI systems capable of safely handling the complexities of real-world clinical data Lambert et al. (2024); Huang et al. (2024); Zou et al. (2023). These methods aim to quantify when a model is uncertain, thereby flagging difficult cases for expert review and avoiding potential misdiagnoses, effectively facilitating a 'second opinion' workflow Kompa et al. (2021). For instance, the Plex framework by Tran et al. evaluates model reliability on retinal datasets by testing for robust generalization under "Country Shift" (a form of covariate shift) and the ability to detect new disease stages in "Severity Shift" scenarios (a semantic shift) Tran et al. (2022). In a more targeted application, Wang et al. developed an Uncertainty-Inspired Open Set (UIOS) model using evidential deep learning to classify nine retinal conditions. Their model assigns a high uncertainty score to out-of-distribution samples (e.g. unseen diseases, low-quality images, or even non-fundus images) prompting a manual check by an ophthalmologist Wang et al. (2023). In the domain of glaucoma, de Vente et al. established the AIROGS benchmark, explicitly designing a challenge to evaluate AI robustness against ungradable fundus images and out-of-distribution samples, emphasizing that clinical reliability hinges on rejecting low-quality inputs de Vente et al. (2024). Similarly, Akram et al. applied Bayesian deep learning to a DenseNet-121 model for diabetic retinopathy classification. By using methods like Monte Carlo Dropout to represent a posterior predictive distribution, their model quantifies predictive uncertainty, which not only improves diagnostic accuracy but also provides crucial confidence estimates for clinical decision-making Akram et al. (2025).

The closest work to ours is the paper by Band et al. Band et al. (2022), which focused only on diabetic retinopathy and evaluated just two datasets, with relatively limited discussion of the risk–coverage–accuracy trade-off. However, they did provide an excellent analysis of thresholding effects and distribution shifts (i.e., country and severity). Therefore, there remains a lack of systematic work evaluating uncertainty quantification (UQ) methods and their advantages and limitations across different aspects of image-based ophthalmic AI, diagnosis, and screening—specifically from the perspective of clinical evaluators and researchers. A comprehensive study is needed to provide practical guidance on calibration analysis and uncertainty considerations. This paper aims to address this gap and provide thorough benchmarking of these methods.

## A.2 Aleatoric and Epistemic Uncertainty

Uncertainty in AI is typically categorized by its source into two main types: **aleatoric** and **epistemic**. Aleatoric uncertainty stems from inherent stochasticity or noise within the data itself and is generally considered irreducible. Epistemic uncertainty, in contrast, originates from the model's limitations, such as being trained on insufficient data, and is related to the limitations in the model itself. It stems from the model's imperfect understanding of the true underlying data distribution and can often be reduced by acquiring more training data or improving the model architecture Wang et al. (2025); Loftus et al. (2022); Gruber et al. (2023).

In the domain of retinal fundus photography and image-based screening, both forms of uncertainty are critically important. Aleatoric uncertainty can arise from sensor noise, motion blur from the patient, or genuine diagnostic ambiguity in cases of early-stage or subtle pathology. Epistemic uncertainty can manifest when a model encounters an image

from an unseen camera type or a rare disease presentation not well-represented in its training data. Models trained on clean, curated public datasets often fail to generalize to the messy, heterogeneous data found in clinical practice (a critical "lab-to-clinic" gap). Without a reliable way to quantify their uncertainty, these models can be confidently wrong, eroding trust among clinicians and posing a risk to patient safety Kompa et al. (2021); Griot et al. (2025).

Table 2: Comparison of types of uncertainty.

| Aleatoric Uncertainty | Epistemic Uncertainty |
|---|---|
| Arises from inherent noise, randomness, or ambiguity in the data itself. | Arises from limitations in the model or insufficient training data. |
| Considered **irreducible**; cannot be reduced by collecting more of the same data. | Considered **reducible**; can be reduced with more diverse data or a better model. |
| **Sources in Fundus Imaging:** Sensor noise, motion artifacts, poor focus, early-stage pathology. | **Sources in Fundus Imaging:** Out-of-distribution data, lack of examples for rare diseases, model misspecification. |
| Quantifies the unpredictability of the system being measured. | Quantifies the model's lack of knowledge about the data-generating function. |

### A.3 UNCERTINATY QUANTIFICATION METHODS

A variety of methods have been proposed to estimate uncertainty in deep neural networks, which can be broadly categorized into several families. In this study, we focus on Post-hoc Methods, which retrofit uncertainty estimation onto pre-trained models without requiring architectural modifications. Table 1 presents the prominent uncertainty quantification (UQ) methods considered in this study along with their details Huang et al. (2024); Lambert et al. (2024); Abdar et al. (2021).

Deterministic Methods produce uncertainty estimates through single forward passes without probabilistic modeling. These methods include Maximum Softmax Probability (MSP), Predictive Entropy, distance-based approaches, ensemble disagreement metrics, and learned uncertainty heads that directly output confidence scores alongside predictions. Test Time Augmentation (TTA) can also be considered a deterministic method when using a fixed set of predefined augmentations (e.g., always flipping). Bayesian Methods treat model parameters as probability distributions rather than point estimates, allowing for the natural capture of epistemic uncertainty, like the Monte Carlo Dropout (MC Dropout) method treats dropout as a Bayesian approximation to variational inference Gal and Ghahramani (2016). It can only be applied post-hoc if the original model was trained with dropout layers. Statistical Methods leverage classical statistical theory, incorporating techniques such as bootstrap sampling for parameter uncertainty, deep ensembles, and conformal prediction, which offers distribution-free coverage guarantees. TTA can be considered a statistical method when augmentations are randomly sampled from probability distributions (e.g., random rotations from a uniform distribution). Deep ensemble is not strictly Bayesian, but empirically approximates Bayesian model averaging and often outperforms more formal Bayesian neural nets Lakshminarayanan et al. (2017). Hybrid Methods combine multiple approaches to leverage their complementary strengths. For example, Bayesian neural networks may utilize deterministic feature extractors, or ensemble methods may incorporate both frequentist and Bayesian components. Deep Ensembles become hybrid when individual ensemble members employ Bayesian techniques (such as MC Dropout) or are combined with other UQ methods. Similarly, test-time augmentation becomes hybrid when the augmentation strategy incorporates learned uncertainty (e.g., learned augmentation policies) or when it is used alongside other uncertainty methods Huang et al. (2024); Lambert et al. (2024); Abdar et al. (2021).

### A.4 List of the used dataset for the benchmarking

Our benchmarking dataset is reported in Table 3. The collected dataset is split in a stratified manner into training (including validation) and test sets. Our in-house local dataset does not provide any training data. Within the test set, 20% of the samples—also selected in a stratified way—are designated as the calibration set, which is used only for calibration and not for evaluation. This was a deliberate methodological choice to reflect a realistic development workflow in which a model is trained in the "lab" but calibrated and evaluated on data from a different distribution—the "clinic." In practice, clinical evaluators only have access to the deployed model and the clinical test population, not the original training or validation data. Therefore, drawing the calibration set from the same distribution as the test set (while keeping it separate) best simulates this lab-to-clinic gap.

Table 3: List of used datasets in this study. The + shows the number of positive samples in that dataset.

| Dataset | Glaucoma | | DR | | AMD | | Test | Note |
|---|---|---|---|---|---|---|---|---|
| | Tot | + | Tot | + | Tot | + | | |
| AIROGS | 101120 | 3270 | | | | | | de Vente et al. (2024) |
| APTOS | | | 3662 | 1857 | | | | Karthik et al. (2019) |
| Aizawl | | | 495 | 452 | | | | Vanlalnunpuia et al. (2025) |
| BRSET | | | 16264 | 1070 | 16264 | 299 | | Nakayama et al. (2024b;a); Goldberger et al. (2000) |
| Cataract | 401 | 101 | | | | | | yiweichen04 (2016) |
| DDR | | | 13585 | 7328 | | | 4074 | Li et al. (2019) |
| DIARETDB1 | | | 89 | 89 | | | | Kauppi et al. (2007) |
| DR1_DR2 | | | 1904 | 972 | | | | Pires et al. (2014) |
| DeepDRiD | | | 1569 | 869 | | | 400 | Liu et al. (2022) |
| Drishti | 101 | 70 | | | | | 51 | Sivaswamy et al. (2014) |
| FIVES | 800 | 200 | 800 | 200 | 800 | 200 | 300 | Jin et al. (2022) |
| G1020 | 1020 | 296 | | | | | | Bajwa et al. (2020) |
| GRAPE | 631 | 631 | | | | | | Huang et al. (2023) |
| IDRID | | | 516 | 348 | | | 103 | Porwal et al. (2018) |
| JICHI | | | 9939 | 3810 | | | | Takahashi et al. (2017) |
| JSIEC | 51 | 13 | 144 | 106 | 38 | | | Cen et al. (2021) |
| KCG | 1450 | 899 | | | | | | Song et al. (2021) |
| LES-AV | 11 | | | | | | | Odstrcilik et al. (2013) |
| Mured | | | 1621 | 322 | 1621 | 131 | 332 | Rodríguez et al. (2022) |
| ODIR | 6985 | 326 | 6985 | 93 | 6985 | 280 | | Wang et al. (2020b) |
| ORIGA | 650 | 168 | | | | | | Zhang et al. (2010) |
| PAPILA | 488 | 155 | | | | | | Kovalyk et al. (2022) |
| RFMiD v1 (RIADD) | | | 2560 | 508 | 2560 | 138 | 640 | Pachade et al. (2021) |
| RFMiD v2 | | | 836 | 70 | 836 | 10 | 167 | Panchal et al. (2023) |
| MESSIDOR2 | | | 1748 | 731 | | | | Abràmoff et al. (2013) |
| SUSTech-SYSU | | | 1219 | 588 | | | | Lin et al. (2020) |
| HYGD | 747 | 548 | | | | | | Abramovich et al.; 2025) |
| TJDR | | | 257 | 257 | | | 55 | Mao et al. (2023) |
| UNA-DR | | | 1437 | 726 | | | | Benítez et al. (2021) |
| e-ophtha | | | 237 | 121 | | | | Decenciere et al. (2013) |
| eyePACS_orig | | | 35125 | 9315 | | | | Cuadros and Bresnick (2009) |
| iChallenge_ADAM | | | | | 400 | 89 | | Fu et al. (2020) |
| iChallenge_GAMMA | 300 | 149 | | | | | 100 | Wu et al. (2023) |
| iChallenge_PALM | | | | | | | | Fang et al. (2024) |
| iChallenge_REFUGE | 1200 | 120 | | | | | 400 | Orlando et al. (2020) |
| mBRSET | | | 4883 | 1134 | | | | Wu et al. (2025) |
| **LOCAL DATASET** | 743 | 365 | | | | | 743 | |

### A.5 Training Protocol

The training dataset is provided via a single CSV file containing labels and paths to the fundus photos. The models are trained for 150 epochs using a *ReduceLROnPlateau* scheduler to dynamically adjust the learning rate, which is halved if the validation loss does not improve for a patience of 10 epochs. 25% of the training set is used as the validation set to monitor the validation loss. Additionally, an early stopping mechanism is employed that terminates training if the validation loss does not improve for 15 consecutive epochs, with the model from the best epoch being saved for inference. All models are trained using the *Adam* optimizer with an initial learning rate of $5 \times 10^{-4}$. To handle class imbalance, we employ a weighted *Cross-Entropy Loss*, where the weights are calculated as the inverse frequency of each class in the training set. A similar effect is observed for ViT (FLSD-53), as it is trained using a weighted focal classification loss. The batch size was set to 250, and the GPU used was a Quadro RTX 6000 with 24 GB of available memory. The entire codebase is implemented in PyTorch.

## A.6 Evaluation Metrics

We employ a comprehensive set of metrics to evaluate predictive performance and uncertainty quality across multiple dimensions.

Classification performance metrics include the Area Under the ROC Curve (AUROC)($\uparrow$), which measures the discriminative ability between healthy and diseased cases across all threshold values, and the Area Under the Precision-Recall Curve (AUPRC)($\uparrow$), which is particularly important for screening applications due to its focus on performance in imbalanced datasets where positive cases are rare. Furthermore: Positive and negative predictive values (PPV and NPV)($\uparrow$) are considered.

For calibration metrics, we use the Expected Calibration Error (ECE($\downarrow$)) to quantify the alignment between predicted confidence and actual accuracy through binned reliability analysis. The Negative Log-Likelihood (NLL($\downarrow$)) assesses the quality of probabilistic predictions, penalizing overconfident incorrect predictions, while the Brier Score ($\downarrow$) evaluates both calibration and sharpness of probabilistic forecasts.

In terms of uncertainty-specific metrics, the Area Under the Risk-Coverage Curve (AURC)($\downarrow$) measures the effectiveness of uncertainty estimates for selective prediction by evaluating risk reduction as coverage decreases. Risk@90%($\downarrow$) Coverage reports the error rate when accepting 90% of samples with the lowest uncertainty, indicating safety at high coverage levels, whereas Coverage@5% ($\uparrow$) Risk determines the fraction of samples that can be processed automatically while maintaining a 5% error rate, which is crucial for clinical deployment.

We also utilize various visualization and analysis methods. ROC curves display the true positive rate versus false positive rate across decision thresholds for each disease and UQ method, while precision-recall curves illustrate the trade-offs between precision and recall, providing important insights for imbalanced medical datasets. Reliability diagrams plot predicted confidence against observed accuracy to visualize calibration quality (closer to the diagonal = better), and risk-coverage curves illustrate the relationship between coverage (the fraction of accepted samples) and risk (error rate) for different uncertainty thresholds. Finally, conformal prediction coverage plots compare nominal versus observed coverage, assessing the statistical validity of uncertainty estimates and revealing systematic under-coverage in raw model predictions.

The reliability diagram and the conformal prediction coverage plot measure different aspects of calibration. The reliability diagram plots average correctness within bins of predicted confidence; if the curve tracks the diagonal, the model is considered well calibrated on average. However, this global view can mask slight overconfidence or underconfidence due to the smoothing effect of binning. In contrast, the conformal prediction coverage plot is more stringent, evaluating how often the chosen confidence level truly covers the correct label. Neural networks often exhibit systematic undercoverage, where high predicted probabilities do not accurately reflect true outcomes. Thus, conformal prediction examines whether prediction sets fulfill statistical guarantees at all coverage levels, indicating that a model could appear reasonably calibrated in one plot but fail in coverage validity tests, especially in the high-confidence range critical for clinical applications.

Together, these metrics provide a thorough assessment of both predictive accuracy and uncertainty reliability, which are essential for determining the readiness of models for clinical deployment.

## A.7 Hyperparameter Selection and Justification

These hyperparameter values were selected to balance statistical robustness with the computational constraints of clinical deployment, grounded in established literature:

- **Monte Carlo Dropout ($T = 50$):** We set the number of stochastic forward passes to $T = 50$. While foundational work by Gal and Ghahramani (2016) suggests that as few as $T = 10$ samples can be sufficient for reasonable uncertainty estimation, we opted for a more conservative value $T = 50 > 10$. The benchmarking study by Band et al. Band et al. (2022) evaluated uncertainty estimation for diabetic retinopathy

using only $T = 5$ Monte Carlo samples. Furthermore, empirical analysis in medical imaging contexts, such as Milanés-Hermosilla et al. (2021), specifically supports $T = 50$ as a "safe choice" where accuracy reaches evident stabilization, ensuring robust convergence of the posterior approximation while maintaining acceptable inference latency for clinical workflows.

- **Deep Ensembles** $(N = 5)$**:** We utilized an ensemble size of $N = 5$. The benchmarking study by Band et al. Band et al. (2022) evaluated uncertainty estimation for diabetic retinopathy using an ensemble size of 3. Our choice $N = 5 > 3$ is also directly supported by the seminal work of Lakshminarayanan et al. (2017) and Ovadia et al. (2019), which demonstrated that an ensemble size of 5 is sufficient to capture the majority of the uncertainty benefit (calibration and accuracy). Increasing $N$ beyond 5 yields diminishing returns that do not justify the linear increase in training and inference costs, a critical consideration for resource-constrained hospital settings.

- **Test-Time Augmentation** $(K = 20)$**:** There is no single consensus on the optimal $K$ in the literature, with values ranging significantly based on the application. Recent retinal and medical imaging studies have utilized values as low as $K = 3$ Itoh et al. (2025) or $K = 4$ Li et al. (2023), while others perform grid searches settling on $K = 6$ Nazzal et al. (2024) or use up to $K = 14$ Yu et al. (2023). We selected $K = 20$ to be on the rigorous end of this spectrum. This choice ensures a low-variance estimation of the predictive distribution Moshkov et al. (2020), prioritizing robustness over the minimal computational savings of smaller $K$ values (e.g., $N = 6$ requires $\approx 2.90$s per image Nazzal et al. (2024)), while acknowledging the linear cost increase.

## A.8 Detailed Tables and Extended Figures

Table 4: Detailed AUROC and AUPRC results (top table) and calibration outcomes (bottom table).

| Disease | ViT | | | | RETFoundGreen | | | | ViT(FLSD-53) | | | |
|---|---|---|---|---|---|---|---|---|---|---|---|---|
| | Baseline | MC Dropout | TTA | Deep Ensemble | Baseline | MC Dropout | TTA | Deep Ensemble | Baseline | MC Dropout | TTA | Deep Ensemble |
| **AUROC** | | | | | | | | | | | | |
| AMD | 0.911 | 0.911 | 0.903 | 0.910 | 0.883 | 0.884 | 0.810 | 0.884 | **0.912** | 0.911 | 0.911 | 0.911 |
| Diabetes | 0.792 | 0.791 | 0.806 | 0.794 | **0.819** | 0.818 | 0.766 | 0.816 | 0.796 | 0.796 | 0.809 | 0.795 |
| Glaucoma | 0.908 | 0.909 | **0.922** | 0.902 | 0.913 | 0.911 | 0.905 | 0.913 | 0.895 | 0.894 | 0.913 | 0.896 |
| **AUPRC** | | | | | | | | | | | | |
| AMD | 0.525 | 0.541 | 0.514 | 0.525 | 0.421 | 0.432 | 0.266 | 0.432 | 0.540 | 0.534 | **0.543** | 0.534 |
| Diabetes | 0.799 | 0.798 | 0.797 | 0.801 | **0.805** | 0.804 | 0.735 | 0.803 | 0.803 | 0.803 | 0.803 | 0.801 |
| Glaucoma | 0.817 | 0.817 | 0.808 | 0.808 | 0.819 | **0.820** | 0.784 | 0.818 | 0.804 | 0.804 | 0.802 | 0.805 |

| Disease | ViT | | | | RETFoundGreen | | | | ViT(FLSD-53) | | | |
|---|---|---|---|---|---|---|---|---|---|---|---|---|
| | Baseline | MC Dropout | TTA | Deep Ens. | Baseline | MC Dropout | TTA | Deep Ens. | Baseline | MC Dropout | TTA | Deep Ens. |
| **NLL** | | | | | | | | | | | | |
| AMD | 0.490 | 0.484 | **0.357** | 0.514 | 0.536 | 0.528 | 0.383 | 0.549 | 0.569 | 0.570 | 0.548 | 0.570 |
| Diabetes | 0.547 | 0.548 | 0.549 | 0.544 | **0.530** | 0.531 | 0.586 | 0.531 | 0.617 | 0.617 | 0.620 | 0.614 |
| Glaucoma | 0.413 | 0.417 | **0.317** | 0.412 | 0.368 | 0.374 | 0.358 | 0.373 | 0.554 | 0.555 | 0.526 | 0.566 |
| **ECE** | | | | | | | | | | | | |
| AMD | 0.257 | 0.264 | 0.183 | 0.270 | 0.276 | 0.282 | **0.173** | 0.288 | 0.350 | 0.350 | 0.336 | 0.350 |
| Diabetes | 0.045 | 0.044 | 0.063 | 0.044 | 0.050 | 0.049 | 0.052 | **0.043** | 0.165 | 0.164 | 0.173 | 0.164 |
| Glaucoma | 0.171 | 0.175 | **0.059** | 0.164 | 0.127 | 0.138 | 0.086 | 0.132 | 0.277 | 0.286 | 0.261 | 0.294 |
| **Brier Score** | | | | | | | | | | | | |
| AMD | 0.156 | 0.155 | **0.112** | 0.164 | 0.172 | 0.171 | 0.118 | 0.177 | 0.190 | 0.191 | 0.180 | 0.190 |
| Diabetes | 0.183 | 0.183 | 0.182 | 0.182 | **0.175** | **0.175** | 0.199 | **0.175** | 0.213 | 0.213 | 0.214 | 0.212 |
| Glaucoma | 0.129 | 0.130 | **0.095** | 0.128 | 0.109 | 0.111 | 0.112 | 0.111 | 0.183 | 0.183 | 0.169 | 0.188 |

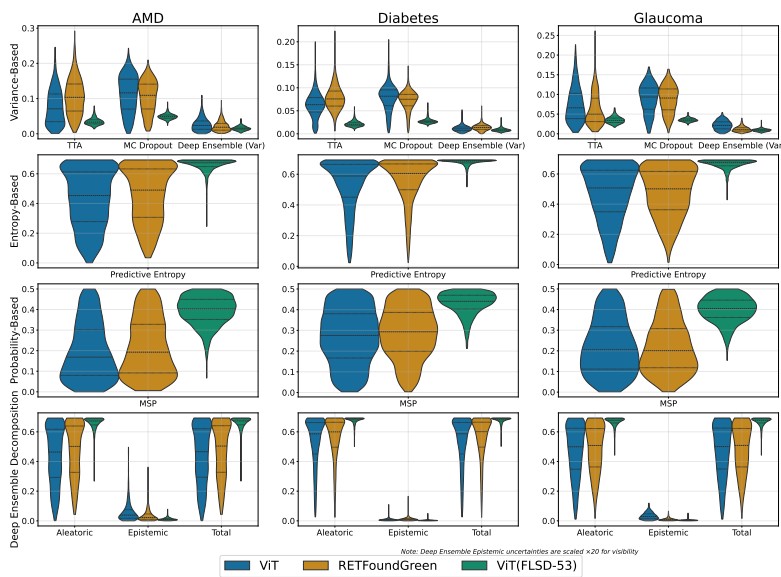

Figure 9: Visualization of the diversity of uncertainty measurements based on diseases and UQ metrics.

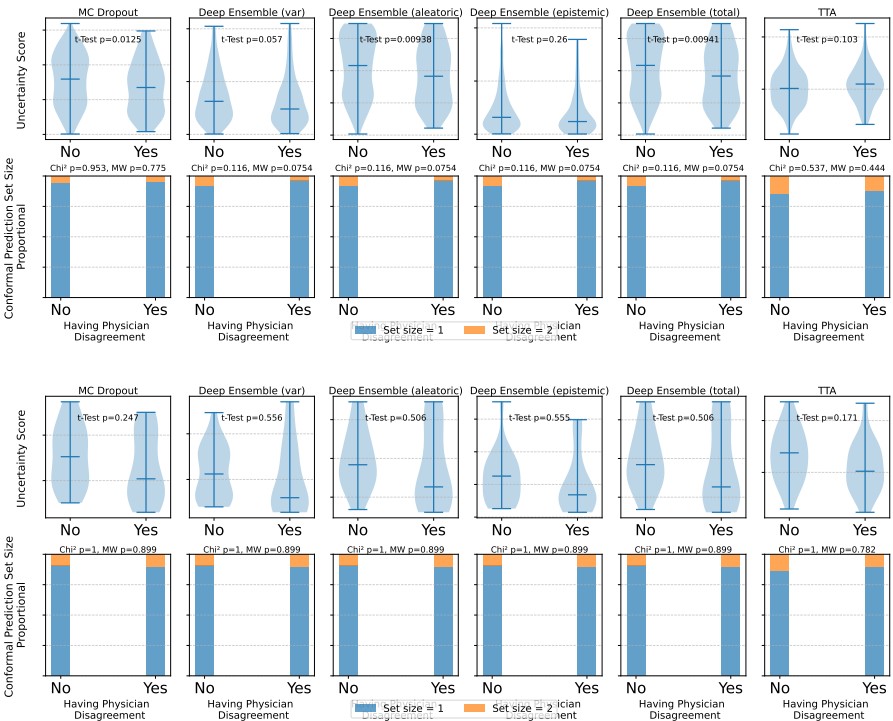

Figure 10: Each subplot presents the results for a different uncertainty quantification method across the disagreement groups (**RETFound-Green model**). Top) Local dataset. Bottom) Drishti dataset.

Table 5: Results of selective prediction investigations analyzing the impact and importance of different UQ methods (top: ViT; middle: RETFound-Green; bottom: ViT(FLSD-53)).

| Disease | Baseline | MSP | Predictive Entropy | MC Dropout | TTA | Deep Ensemble Total | Aleatoric | Epistemic | Var |
|---|---|---|---|---|---|---|---|---|---|
| **AURC** | | | | | | | | | |
| AMD | 0.407 | 0.126 | 0.126 | 0.128 | 0.052 | 0.142 | 0.142 | 0.172 | 0.154 |
| Diabetes | 0.211 | 0.167 | 0.167 | 0.178 | 0.184 | 0.165 | 0.165 | 0.216 | 0.190 |
| Glaucoma | 0.224 | 0.078 | 0.078 | 0.078 | 0.042 | 0.078 | 0.078 | 0.132 | 0.106 |
| **Risk @ 90% Cov** | | | | | | | | | |
| AMD | 0.232 | 0.186 | 0.186 | 0.193 | 0.122 | 0.197 | 0.194 | 0.210 | 0.207 |
| Diabetes | 0.270 | 0.245 | 0.245 | 0.258 | 0.250 | 0.242 | 0.242 | 0.257 | 0.252 |
| Glaucoma | 0.186 | 0.145 | 0.145 | 0.146 | 0.108 | 0.141 | 0.140 | 0.152 | 0.144 |
| **Cov @ 5% Risk** | | | | | | | | | |
| AMD | 0.002 | 0.047 | 0.047 | 0.023 | 0.573 | 0.021 | 0.021 | 0.021 | 0.022 |
| Diabetes | 0.040 | 0.036 | 0.036 | 0.053 | 0.016 | 0.040 | 0.040 | 0.005 | 0.023 |
| Glaucoma | 0.074 | 0.437 | 0.437 | 0.360 | 0.608 | 0.385 | 0.385 | 0.038 | 0.175 |

| Disease | Baseline | MSP | Predictive Entropy | MC Dropout | TTA | Deep Ensemble Total | Aleatoric | Epistemic | Var |
|---|---|---|---|---|---|---|---|---|---|
| **AURC** | | | | | | | | | |
| AMD | 0.447 | 0.162 | 0.162 | 0.152 | 0.072 | 0.172 | 0.172 | 0.195 | 0.181 |
| Diabetes | 0.210 | 0.153 | 0.153 | 0.166 | 0.236 | 0.155 | 0.155 | 0.212 | 0.194 |
| Glaucoma | 0.171 | 0.055 | 0.055 | 0.056 | 0.053 | 0.057 | 0.057 | 0.090 | 0.071 |
| **Risk @ 90% Cov** | | | | | | | | | |
| AMD | 0.252 | 0.210 | 0.210 | 0.210 | 0.142 | 0.220 | 0.219 | 0.233 | 0.230 |
| Diabetes | 0.258 | 0.222 | 0.222 | 0.236 | 0.286 | 0.225 | 0.224 | 0.252 | 0.243 |
| Glaucoma | 0.139 | 0.098 | 0.098 | 0.104 | 0.134 | 0.100 | 0.101 | 0.116 | 0.116 |
| **Cov @ 5% Risk** | | | | | | | | | |
| AMD | 0.001 | 0.003 | 0.003 | 0.014 | 0.393 | 0.004 | 0.004 | 0.000 | 0.004 |
| Diabetes | 0.011 | 0.011 | 0.011 | 0.014 | 0.001 | 0.009 | 0.009 | 0.005 | 0.029 |
| Glaucoma | 0.055 | 0.497 | 0.497 | 0.515 | 0.589 | 0.517 | 0.519 | 0.108 | 0.317 |

| Disease | Baseline | MSP | Predictive Entropy | MC Dropout | TTA | Deep Ensemble Total | Aleatoric | Epistemic | Var |
|---|---|---|---|---|---|---|---|---|---|
| **AURC** | | | | | | | | | |
| AMD | 0.402 | 0.122 | 0.122 | 0.168 | 0.132 | 0.126 | 0.126 | 0.171 | 0.165 |
| Diabetes | 0.211 | 0.163 | 0.163 | 0.250 | 0.267 | 0.164 | 0.164 | 0.274 | 0.269 |
| Glaucoma | 0.209 | 0.072 | 0.072 | 0.120 | 0.116 | 0.081 | 0.081 | 0.177 | 0.171 |
| **Risk @ 90% Cov** | | | | | | | | | |
| AMD | 0.230 | 0.179 | 0.179 | 0.195 | 0.179 | 0.182 | 0.187 | 0.203 | 0.201 |
| Diabetes | 0.272 | 0.238 | 0.238 | 0.264 | 0.262 | 0.239 | 0.239 | 0.265 | 0.263 |
| Glaucoma | 0.173 | 0.132 | 0.132 | 0.137 | 0.129 | 0.139 | 0.143 | 0.164 | 0.157 |
| **Cov @ 5% Risk** | | | | | | | | | |
| AMD | 0.003 | 0.060 | 0.060 | 0.003 | 0.013 | 0.050 | 0.050 | 0.014 | 0.021 |
| Diabetes | 0.060 | 0.052 | 0.052 | 0.001 | 0.000 | 0.042 | 0.042 | 0.000 | 0.000 |
| Glaucoma | 0.089 | 0.394 | 0.394 | 0.003 | 0.038 | 0.313 | 0.313 | 0.002 | 0.002 |

Table 6: Observed coverage with and without conformal prediction (coverage 90%) across diseases and methods.

|          | None | MC Dropout | Deep Ensemble | TTA |
|----------|------|------------|---------------|-----|
| **Observed Coverage (CP) (%)** | | | | |
| AMD | 89.91 | 86.78 | 89.30 | 90.31 |
| Diabetes | 90.31 | 89.27 | 90.27 | 90.18 |
| Glaucoma | 89.21 | 88.70 | 90.92 | 87.50 |
| **Observed Coverage (Raw) (%)** | | | | |
| AMD | 28.15 | 23.11 | 26.54 | 36.33 |
| Diabetes | 12.82 | 11.80 | 12.52 | 12.27 |
| Glaucoma | 20.72 | 19.01 | 21.92 | 37.33 |

ViT

|          | None | MC Dropout | Deep Ensemble | TTA |
|----------|------|------------|---------------|-----|
| **Observed Coverage (CP) (%)** | | | | |
| AMD | 90.21 | 90.72 | 89.10 | 91.62 |
| Diabetes | 91.99 | 90.99 | 89.46 | 89.63 |
| Glaucoma | 90.58 | 91.10 | 92.12 | 85.96 |
| **Observed Coverage (Raw) (%)** | | | | |
| AMD | 23.51 | 19.68 | 20.48 | 19.88 |
| Diabetes | 8.26 | 7.69 | 8.16 | 2.43 |
| Glaucoma | 19.18 | 15.75 | 17.64 | 36.99 |

RETFoundGreen

|          | None | MC Dropout | Deep Ensemble | TTA |
|----------|------|------------|---------------|-----|
| **Observed Coverage (CP) (%)** | | | | |
| AMD | 90.92 | 90.21 | 89.71 | 88.19 |
| Diabetes | 90.10 | 90.20 | 89.01 | 91.08 |
| Glaucoma | 87.67 | 92.12 | 95.21 | 90.07 |
| **Observed Coverage (Raw) (%)** | | | | |
| AMD | 0.20 | 0.10 | 0.10 | 0.10 |
| Diabetes | 0.00 | 0.00 | 0.00 | 0.00 |
| Glaucoma | 0.00 | 0.00 | 0.00 | 0.00 |

ViT(FLSD-53)

Table 7: Detection and Calibration performance on local dataset (glaucoma only).

| | ViT | | | | RETFoundGreen | | | | ViT(FLSD-53)) | | | |
|---|---|---|---|---|---|---|---|---|---|---|---|---|
| Metric | Baseline | MC Dropout | TTA | Deep Ensemble | Baseline | MC Dropout | TTA | Deep Ensemble | Baseline | MC Dropout | TTA | Deep Ensemble |
| AUROC | 0.778 | 0.779 | 0.742 | 0.773 | 0.690 | 0.691 | 0.670 | 0.671 | 0.767 | 0.771 | 0.749 | 0.769 |
| AUPRC | 0.743 | 0.745 | 0.733 | 0.745 | 0.672 | 0.674 | 0.648 | 0.664 | 0.737 | 0.734 | 0.723 | 0.739 |

| | ViT | | | | RETFoundGreen | | | | ViT(FLSD-53)) | | | |
|---|---|---|---|---|---|---|---|---|---|---|---|---|
| Metric | Baseline | MC Dropout | TTA | Deep Ensemble | Baseline | MC Dropout | TTA | Deep Ensemble | Baseline | MC Dropout | TTA | Deep Ensemble |
| NLL | 0.769 | 0.748 | 0.651 | 0.738 | 0.956 | 0.927 | 0.670 | 0.984 | **0.640** | **0.640** | 0.646 | 0.643 |
| ECE | 0.252 | 0.243 | 0.143 | 0.233 | 0.322 | 0.314 | **0.071** | 0.331 | 0.142 | 0.144 | 0.130 | 0.149 |
| Brier | 0.263 | 0.258 | 0.228 | 0.252 | 0.324 | 0.318 | 0.236 | 0.333 | **0.224** | 0.224 | 0.227 | 0.226 |

Table 8: Selective prediction metrics (AURC, Risk@90% Cov, Cov@5% Risk) on the Glaucoma local test set. The best performance for each metric is highlighted in bold.

| Metric | Baseline | MSP | Predictive Entropy | MC Dropout | TTA | Deep Ensemble (Total) | Aleatoric | Epistemic | Var |
|---|---|---|---|---|---|---|---|---|---|
| **ViT** | | | | | | | | | |
| AURC | 0.316 | 0.289 | 0.289 | 0.291 | 0.282 | **0.281** | **0.281** | 0.305 | 0.290 |
| Risk @ 90% Cov | 0.449 | 0.400 | 0.400 | 0.392 | **0.371** | 0.382 | 0.379 | 0.393 | 0.392 |
| Cov @ 5% Risk | 0.009 | 0.009 | 0.009 | 0.009 | 0.009 | 0.012 | 0.012 | **0.030** | 0.021 |
| **RETFoundGreen** | | | | | | | | | |
| AURC | 0.366 | 0.361 | 0.361 | 0.357 | **0.326** | 0.368 | 0.368 | 0.335 | 0.338 |
| Risk @ 90% Cov | 0.483 | 0.463 | 0.463 | 0.455 | **0.376** | 0.472 | 0.475 | 0.434 | 0.447 |
| Cov @ 5% Risk | 0.007 | 0.007 | 0.007 | 0.007 | 0.019 | 0.009 | 0.009 | **0.021** | 0.012 |
| **ViT(FLSD-53)** | | | | | | | | | |
| AURC | 0.309 | **0.270** | **0.270** | 0.349 | 0.354 | 0.276 | 0.276 | 0.338 | 0.331 |
| Risk @ 90% Cov | 0.395 | **0.348** | **0.348** | 0.356 | 0.380 | 0.371 | 0.367 | 0.395 | 0.395 |
| Cov @ 5% Risk | 0.009 | 0.009 | 0.009 | **0.016** | 0.012 | 0.009 | 0.009 | 0.012 | 0.014 |

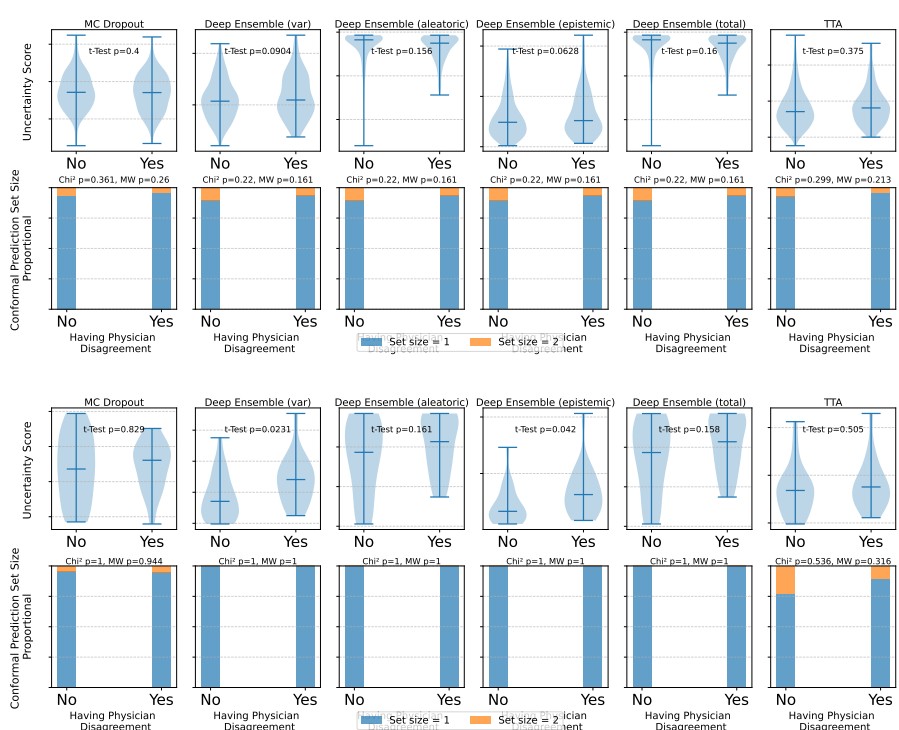

Figure 11: Each subplot presents the results for a different uncertainty quantification method across the disagreement groups (**ViT(FLSD-53)**). Top) Local dataset. Bottom) Drishti dataset.

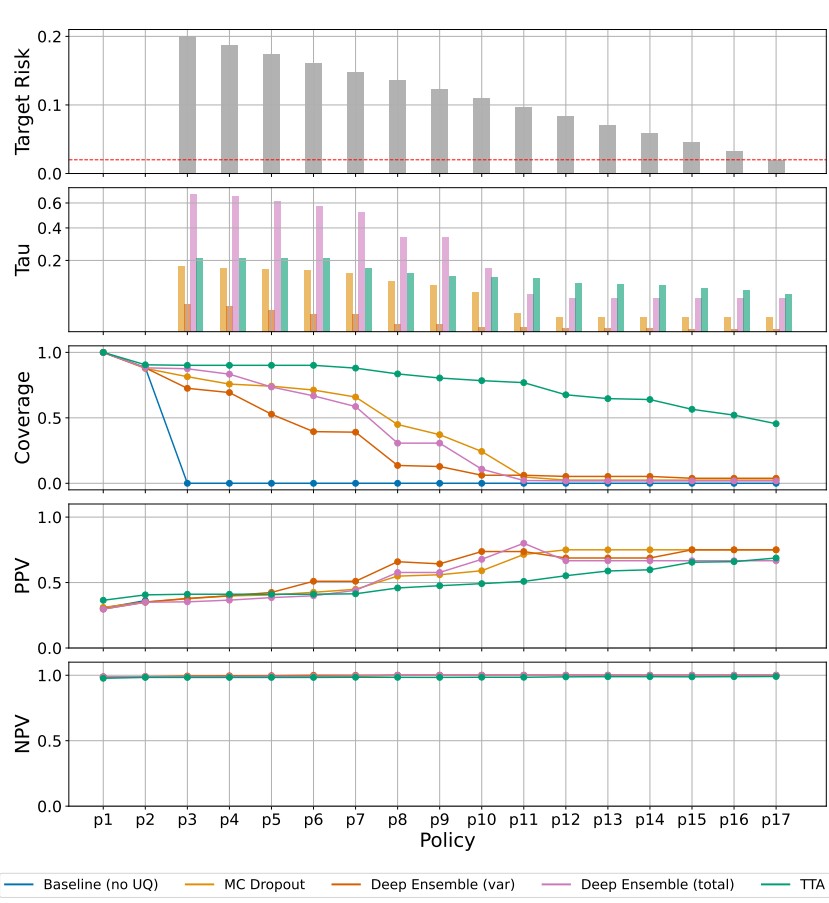

Figure 12: Clinical task target analysis compares **ViT** with several UQ methods across a sweep of acceptable risk targets (p3–p17; p1 = baseline, p2 = conformal filtering with $\alpha = 0.1$). For each target, a threshold $\tau$ on the uncertainty score is chosen to meet that risk, and the resulting coverage (auto-accepted fraction), PPV, and NPV are reported. **AMD**

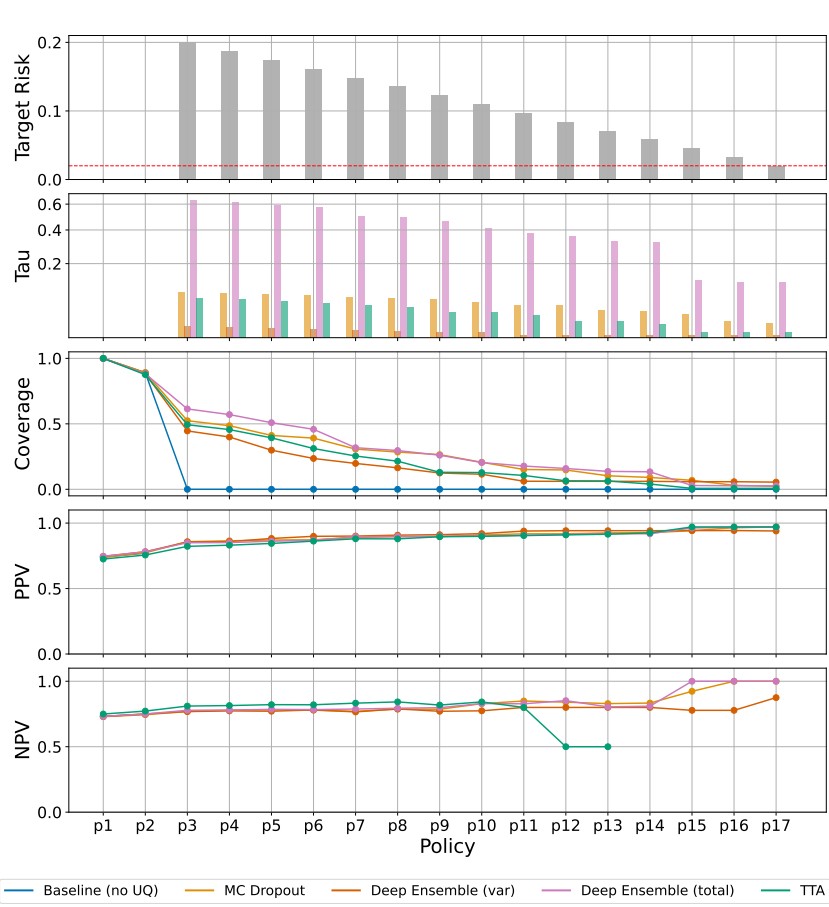

Figure 13: Clinical task target analysis compares **ViT** with several UQ methods across a sweep of acceptable risk targets (p3–p17; p1 = baseline, p2 = conformal filtering with $\alpha = 0.1$). For each target, a threshold $\tau$ on the uncertainty score is chosen to meet that risk, and the resulting coverage (auto-accepted fraction), PPV, and NPV are reported. **Diabetics**

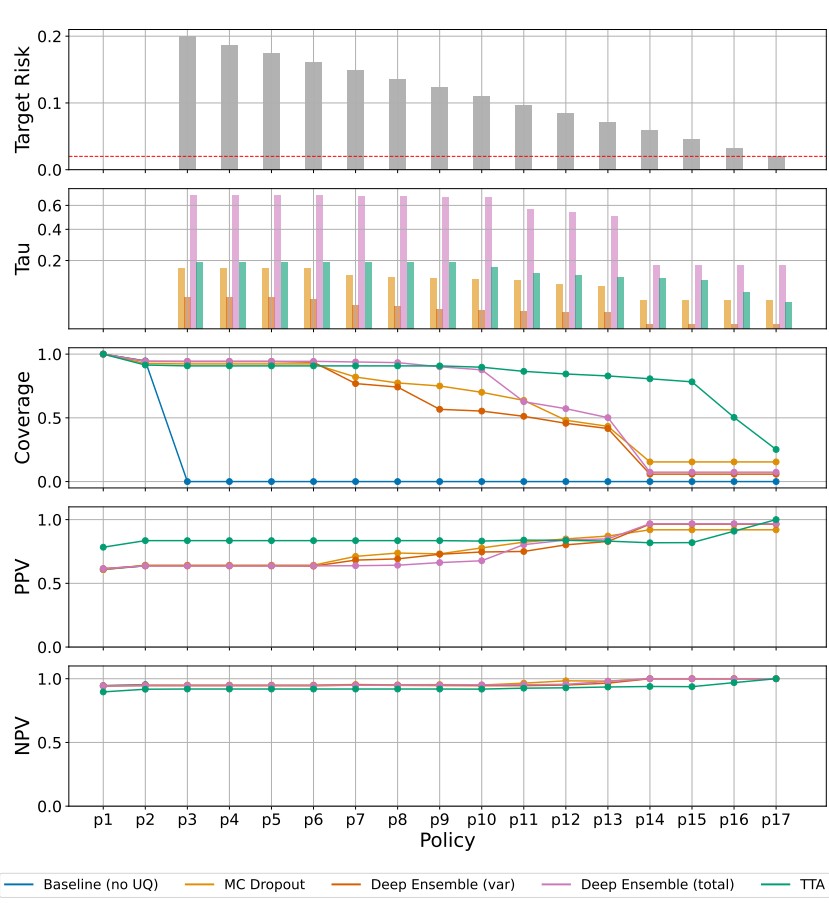

Figure 14: Clinical task target analysis compares **ViT** with several UQ methods across a sweep of acceptable risk targets (p3–p17; p1 = baseline, p2 = conformal filtering with $\alpha = 0.1$). For each target, a threshold $\tau$ on the uncertainty score is chosen to meet that risk, and the resulting coverage (auto-accepted fraction), PPV, and NPV are reported. **Glaucoma**

