# OpenReview forum: "Uncertainty quantification in clinical settings: A retinal fundus screening study and benchmarking"
_ICLR.cc/2026/Conference — Submitted to ICLR 2026_

### Official Review · Reviewer_gA5j · 2025-10-29

**Soundness:** 3
**Presentation:** 3
**Contribution:** 3
**Rating:** 8
**Confidence:** 3

**Summary:**

The article studies classification performance and uncertainty quantification in a retinal fundus screening. The study is based on two vision transformers that are pretrained on ImageNet or on a large dataset of unlabeled fundus images, and evaluates these models both on publicly available datasets (~10-100K images) as well as a separate clinical dataset collected at a local hospital (500 images). Different uncertainty quantification strategies are evaluated and compared to ground-truth test data. Key findings are i) there is a significant drop in classification performance from the public dataset to the private one, raising doubts about the applicability of current machine learning models in clinical practice, ii) uncertainty quantification approaches generally improve reliability, albeit only to a small degree and not consistent with uncertainty patterns reflected in clinicians, iii) split-conformal prediction is essential for callibration, as model confidences and uncertainty estimates tend to be severly overconfident.

**Strengths:**

This is a practical article that provides a summary of the status quo in uncertainty quantification and deep learning in a clinically relevant use-case and with actual data from real-world-deployment (lab to clinic). The article is well-written and the research is sound and carried out well. I very much enjoyed reading.

The fact that lab-to-clinic experiments are made makes the contributions significant and credible. The article has the potential to guide future research in uncertainty quantification and retinal fundus screening and has therefore significant scientific value. I recommend acceptance.

**Weaknesses:**

There are very few weak points that I could identify (see also questions). The article's contribution is not a new methodology but a scientifically sound evaluation of the status quo, which is largely missing in todays AI landscape, so the lack of new methodology should not be counted as weakness.

A weakness appears to be the focus on vision transformer architectures, which tend to be really large machine learning models that require substantial amounts of pretraining. It is unclear whether smaller model architectures would have performed better in this setting, as datasets are relatively small. However, the approaches are representative for the status quo in deep learning.

The real-world dataset for testing real-world-deployment appears to have a comparable small size (500 instances) from a machine learning perspective.

**Questions:**

Will all models and datasets be publicly released?

Can the authors further motivate the focus on vision transformers for their study?

The discrepency between model/UQ performance on publicly sourced data and newly collected data is striking to me. How much would a little bit of fine tuning on test-examples help with model performance? Can the authors pinpoint the reasons for this significant drop (e.g. preprocessing of images, different measurement device, etc.)?

---

> ### Author Response · Authors · 2025-11-24
> **Detailed Response**
>
> ### Comment Part1:
> A weakness appears to be the focus on vision transformer architectures, which tend to be really large machine learning models that require substantial amounts of pretraining. It is unclear whether smaller model architectures would have performed better in this setting, as datasets are relatively small. However, the approaches are representative of the status quo in deep learning.
>
> ### Response Part1:
>
> Many thanks for considering our manuscript and for finding our work valuable and impactful.
> Our selection of ViT is motivated by recent trends in foundation models, including the availability of retina-specific fundus foundation models trained on millions of retinal images. Accordingly, we selected RETFoundGreen as the foundation model, along with a DINOv2-pretrained ViT for comparison. In the revised version, we also included ViT (FLSD-53), which is the same ViT architecture trained with focal loss—a method shown in the literature to yield better calibration performance.
> Your point about smaller networks is also valid. However, in addition to recent SOTA models such as the RETFound foundation model, our networks (while large) require training only the final layer with the backbone frozen, resulting in roughly 700 parameters to learn.
>
> ---
>
> ### Comment Part2:
> The real-world dataset for testing real-world deployment appears to have a comparatively small size (500 instances) from a machine learning perspective.
>
> ### Response Part2:
>
> You are right. Creating multi-rater clinical datasets is time-consuming, and unfortunately, it is not always the top priority for physicians. However, we expanded our dataset by an additional 50%, and our local dataset now includes approximately 743 photos. All figures and results have been updated accordingly, with only minor changes in the overall findings and conclusions.
> Our local dataset is still relatively small, but we believe it is now suitable and acceptable as an external clinical dataset. In addition, it is a multi-rater dataset, which allowed us to analyze aleatoric uncertainty arising from physician disagreement.
>
>
> ---
>
> ### Comment Part3:
> Will all models and datasets be publicly released?
>
> ### Response Part3:
>
> Yes, all the trained models (45 total across 3 diseases, 5 ensembles, and 3 DL models) will be shared on HuggingFace, along with the training, inference, and calibration/uncertainty analysis code on GitHub.
> The CSV files for the training, calibration, and test sets—with labels—will also be released as part of the benchmark, and other researchers can download all dataset images themselves.
>
>
> ---
> ### Comment Part4:
> Can the authors further motivate the focus on vision transformers for their study?
>
> ### Response Part4:
>
> As described above, in short: because the available retina foundation model is based on ViT, we considered its variations to stay aligned with the state of the art in fundus image analysis.
>
> ---
> ### Comment Part5:
> The discrepancy between model/UQ performance on publicly sourced data and newly collected data is striking. How much would fine-tuning help? Can the authors pinpoint causes for the drop (e.g., preprocessing, device differences)?
>
> ### Response Part5:
>
> Very good observation. We considered your question and performed fine-tuning on a subset of our local dataset containing 100 photos (after pretraining on publicly available datasets). This improved AUROC by approximately 10%, but the performance was still about 10% lower than the in-distribution results from the public datasets.
>
> We did not investigate the exact cause, but several factors may contribute. The field of view in our images is similar to the other datasets, and we used only high-quality images with identical preprocessing. All of our local images were acquired using a single device (Zeiss Visucam), which is widely used. However, because many public datasets do not report their acquisition devices, we cannot confirm whether this model was represented in the training set—though given its popularity, we believe it is unlikely that it was entirely absent.
> Therefore, we suspect that demographic differences, as well as aleatoric uncertainties such as differences in grading guidelines and pathology scoring, may be contributing factors.
>
>
> ---
> We hope our answers justify a higher score and respectfully ask you to reconsider our rating.
> Thank you, and please let us know if any other comments or concerns remain.

---

> > ### Comment · Reviewer_gA5j · 2025-11-26
> > **Rebuttal Acknowledgment**
> >
> > Dear authors,
> >
> > thank you very much for the informative response and for improving the manuscript further (along some of the points mentioned). The rebuttal addressed all my points and I will vote for acceptance.

---

### Official Review · Reviewer_y3sp · 2025-10-31

**Soundness:** 2
**Presentation:** 3
**Contribution:** 1
**Rating:** 2
**Confidence:** 4

**Summary:**

This paper presents a study of a real benchmark that provides reality check on the promises of medical AI.
The authors used real-world clinical dataset to measure the crucial "lab-to-clinic" gap. Their findings are interesting; we are still a long way from safely automating these screening tasks. This is a good message for the community. Plus, they deserve credit for open-sourcing everything: their code, models, and data splits which sets a high bar for reproducibility and makes their work useful.
That said, my main concern is that they put all their eggs in one basket by only using Vision Transformer models. I'm left wondering if these same conclusions hold true for the classic CNNs that many people still use. It feels like a missed opportunity to make their findings more universal. They also completely sidestepped the practical cost of these methods. A "Deep Ensemble" sounds great, but it means training and running five models instead of one, which is a massive resource drain. For a hospital on a budget, that’s a potential deal-breaker. A bit of discussion on this trade-off would have made the paper more grounded.

**Strengths:**

On top of the points in the summary, these also are strengths in the paper
•	Comprehensive benchmark covering three diseases with rigorous evaluation metrics.
•	Strong commitment to reproducibility (releasing code, models, splits, demo).

**Weaknesses:**

Many issues with the paper in its current form:
•	Limited Novelty: The level of mathematical grounding of the ideas in the paper is somewhat below ICLR standards. The ICLR standards assume theoretical mathematical analysis of “why” these methods fails, in addition to the empirical evidence.
•	A main concern is that they put all their eggs in one basket by only using Vision Transformer models. I'm left wondering if these same conclusions hold true for the classic CNNs that many people still use.
•	Small clinical dataset (536 images).
•       Only two ViT architectures tested, both with frozen features.
•	AMD models perform poorly (low AUPRC).
•	Benchmarking study with no methodological novelty.
•	No justification for hyperparameters (T=50, N=5, K=20).
•	Identifies domain shift degradation but offers little insight into why or how to fix it

**Questions:**

see above

---

> ### Author Response · Authors · 2025-11-24
>
> ## Comment Part1:
> • Limited Novelty: The level of mathematical grounding of the ideas in the paper is somewhat below ICLR standards. The ICLR standards assume theoretical mathematical analysis of “why” these methods fail, in addition to empirical evidence.
> • Benchmarking study with no methodological novelty.
>
> ## Response Part1:
>
> Thanks for reading our work and providing comments.
> Respectfully, let us clarify our work objectives.
> **Our goal was to highlight the importance of this topic and to provide clear guidance for clinical regulators and researchers on how to evaluate AI models in terms of risk–coverage–accuracy.**
> Therefore, our study focuses intentionally on a **post-hoc** evaluation setting, and we addressed it comprehensively with the largest dataset and benchmarking, as well as an in-house clinical dataset.
>
> Respectfully, we do not agree with the comment regarding mathematical analysis and the novelty of our benchmarking contribution. For reference, please consider the following benchmarking papers published in ICLR (<https://openreview.net/forum?id=VvDEuyVXkG>, <https://openreview.net/forum?id=MKEHCx25xp>) and NeurIPS (<https://datasets-benchmarks-proceedings.neurips.cc/paper/2021/file/ac1dd209cbcc5e5d1c6e28598e8cbbe8-Paper-round2.pdf>) where the novelty lies primarily in benchmarking, discussion, and sharing repositories/models.
>
> Our contribution is not a new theoretical proof, but **a large-scale “reality check” of existing theoretical methods in a high-stakes clinical environment**. Evaluating the gap between theoretical assumptions and real-world behavior is, in our view, a critical scientific contribution.
>
> ---
>
> ## Comment Part2:
> • A main concern is the exclusive use of Vision Transformers; would conclusions differ for CNNs?
>
> ## Response Part2:
>
> Very good observation and fair question.
> Our selection of ViTs is motivated by recent trends in foundation models, including the availability of retina-specific fundus foundation models trained on millions of retinal images—all ViT-based. These models consistently outperform CNN-based architectures in the literature. Accordingly, we selected RETFoundGreen as the foundation model, along with a DINOv2-pretrained ViT for comparison. In the revised version, we also included a ViT (FLSD-53) model trained with focal loss.
>
> ---
>
> ## Comment Part3:
>
> • Small clinical dataset (536 images).
>
> ## Response Part3:
>
> You are right. Creating multi-rater clinical datasets is time-consuming and not always the top priority for physicians. However, we expanded our dataset by ~50%, and our local dataset now includes approximately 743 photos. All figures and results have been updated accordingly with only minor changes to conclusions.
> Although relatively small, our dataset is comparable to or larger than existing benchmark glaucoma datasets (PAPILA 488, ORIGA 650, GRAPE 631, DRISHTI 101). Additionally, it is multi-rater, enabling analysis of aleatoric uncertainty from physician disagreement.
>
> ---
> ## Comment Part4:
>
> • Only two ViT architectures tested, both with frozen features.
>
> ## Response Part4:
>
> A fair point. However, training only the classifier head is common practice. Our objective was to analyze risk–coverage–accuracy behavior for clinical evaluators, and we showed that even high-AUROC/AUPRC models can still have major reliability issues.
> While this setup has limitations, we believe it provides an essential baseline. We agree that future work should explore additional architectures and uncertainty-aware training strategies.
>
> **(the rest of the responses in the next official comment: Below)**

---

> > ### Author Response · Authors · 2025-11-24
> >
> > ## Comment Part5:
> > • AMD models perform poorly (low AUPRC).
> >
> > ## Response Part5:
> >
> > The purpose of a benchmark is to report the *actual* state of the field, not to demonstrate optimal performance. The low AUPRC for AMD is therefore an important and novel finding, highlighting a gap that the research community must address.
> >
> > ---
> > ## Comment Part6:
> > • No justification for hyperparameters (T=50, N=5, K=20).
> >
> > ## Response Part6:
> >
> > We thank the reviewer for the request. We added detailed justification in the appendix. These values grounded in the literature (referecnes are available in the general response section above):
> >
> > - Monte Carlo Dropout (T = 50): We set the number of stochastic forward passes to T = 50.
> > While foundational work by [2] suggests that as few as T = 10 samples can be sufficient forreasonable uncertainty estimation, we opted for a more conservative value T = 50 > 10. The benchmarking study by Band et al. [1] evaluated uncertainty estimation for diabetic retinopathy using only T = 5 Monte Carlo samples. Furthermore, empirical analysis in medical imaging contexts, such as [6], specifically supports T = 50 as a “safe choice” where accuracy reaches evident stabilization, ensuring robust convergence of the posterior approximation while maintaining acceptable inference latency for clinical workflows.
> >
> > - Deep Ensembles (N = 5): We utilized an ensemble size of N = 5. The benchmarking study
> > by Band et al. [1] evaluated uncertainty estimation for diabetic retinopathy using an ensemble size of 3. Our choice N = 5 > 3 is also directly supported by the seminal work of [4] and [9], which demonstrated that an ensemble size of 5 is sufficient to capture the majority of the uncertainty benefit (calibration and accuracy). Increasing N beyond 5 yields diminishing returns that do not justify the linear increase in training and inference costs, a critical consideration for resource-constrained hospital settings.
> >
> > - Test-Time Augmentation (K = 20): There is no single consensus on the optimal K in the
> > literature, with values ranging significantly based on the application. Recent retinal and medical imaging studies have utilized values as low as K = 3 [3] or K = 4 [5], while others perform grid searches settling on K = 6 [8] or use up to K = 14 [12]. We selected K = 20 to be on the rigorous end of this spectrum. This choice ensures a low-variance estimation of the predictive distribution [7], prioritizing robustness over the minimal computational savings of smaller K values (e.g., N = 6 requires ≈ 2.90s per image [8]), while acknowledging the linear cost increase
> >
> > ---
> >
> > ## Comment Part7:
> > • Identifies domain-shift degradation but lacks explanation or mitigation strategies.
> >
> > ## Response Part7:
> >
> > Thank you for the great suggestion. We have added this section in the revised version:
> >
> > It may be because the training data do not fully represent (not limited to) the hospital’s imaging devices (e.g., Topcon, Zeiss), variations in sensor resolution, dynamic range, illumination, flash intensity, and color calibration, as well as differences in patient demographics (e.g., age, race), disease presentations
> > (e.g., under- or overrepresentation of mild cases), or clinical workflows (e.g., with or without dilation).
> > These factors introduce significant domain shifts that reduce model generalization. The amount of degradation may be reduced by considering domain-shift adaptation techniques [13], such as test-time adaptation methods (e.g., TENT [11] or TTT [10]), where the model dynamically updates its normalization statistics or minimizes prediction entropy on incoming test streams.
> >
> >
> > ---
> >
> > ## Summary of our response:
> >
> > We again thank you for reading our paper carefully and providing constructive comments. We respectfully disagree that mathematical novelty is required for a benchmarking submission at ICLR, as demonstrated by the references we provided. Based on your feedback, we expanded our in-house dataset, clarified the objectives of the paper, added a detailed hyperparameter justification section, and included an extended discussion on domain shift. We believe that our thorough responses and the substantial improvements to the manuscript merit a higher score, and we kindly ask that you consider revising our rating. We appreciate your efforts/time, and please let us know if any additional comments or concerns remain.

---

### Official Review · Reviewer_Ug1G · 2025-10-31

**Soundness:** 3
**Presentation:** 4
**Contribution:** 3
**Rating:** 8
**Confidence:** 3

**Summary:**

This paper proposes a large-scale benchmark for uncertainty quantification in retinal AI screening across three diseases glaucoma, diabetic retinopathy, and age-related macular degeneration. They evaluate six post-hoc UQ techniques on two ViT backbones (ViT/DINOv2 and RETFound-Green). The authors assemble >114k glaucoma, >100k DR, and >28k AMD fundus images; define standardized train/test/calibration splits; and report discrimination, calibration, selective prediction, and conformal prediction (CP) results, including an external clinical test on a 536-image glaucoma set with 3-rater labels.

**Strengths:**

Clear, end-to-end benchmark framing. The work covers discrimination (AUROC/AUPRC), calibration (ECE/NLL/Brier), selective prediction (AURC, Risk@90% coverage, Coverage@5% risk), and CP validity, with sound descriptions of each metric.
Risk–coverage analyses are thoughtfully interpreted; ensembles show the most consistent gains in glaucoma/AMD, while signals are weaker for DR.
External clinical test & disagreement analysis. The hospital glaucoma set (n=536) reveals a realistic domain shift; the analysis of physician disagreement vs. UQ scores (with a significance test) is valuable.

**Weaknesses:**

Missing baselines and UQ methods post-hoc calibration baselines like temperature scaling and isotonic regression/Dirichlet calibration are not compared. Similarly, Laplace approximation, SWAG, and evidential deep learning

**Questions:**

Why was the calibration set drawn from the test sets rather than from held-out training/validation data?
it would be also useful if the authors provides reliability diagrams (before/after best calibrator) with ECE bins fixed across methods.

---

> ### Author Response · Authors · 2025-11-24
>
> ## Comment Part1:
>
> Missing baselines and UQ methods: post-hoc calibration baselines like temperature scaling and isotonic regression/Dirichlet calibration are not compared. Similarly, Laplace approximation, SWAG, and evidential deep learning.
>
> ## Response Part2:
> Many thanks for considering our manuscript and for finding our work valuable and impactful.
>
> _Our goal was to **highlight the importance of this topic and to provide clear guidance for clinical researchers/evaluators on how to evaluate AI models in terms of risk–coverage–accuracy.**
> In real clinical deployment settings, regulators and clinical researchers do not retrain, modify, or adjust model parameters. As a result, clinical evaluators typically test vendor-provided models as-is, without any modifications.
> Laplace approximation, SWAG, and evidential deep learning all require retraining or access to model parameters, which places them outside our strict **post-hoc** evaluation scope. We have revised the manuscript to explicitly clarify this constraint.
>
> That said, your suggestion regarding additional post-hoc calibration methods is appreciated. Temperature scaling, isotonic regression, and Dirichlet calibration indeed fall within the permissible post-hoc category. However, in clinical settings, calibration is typically the developer’s responsibility rather than the clinician’s, and hospitals do not calibrate vendor-provided AI models. Therefore, **while we emphasize that reliability diagrams are valuable tools that clinical evaluators should examine, performing the calibration itself was outside the primary scope of our study.**
>
> However, we appreciate the importance of these methods and have **now included** the classification focal-loss training paradigm (FLSD-53), which is an uncertainty-aware and calibration-oriented method that may improve calibration.
> (NeurIPS 2020: *Calibrating Deep Neural Networks using Focal Loss*: <https://papers.neurips.cc/paper/2020/file/aeb7b30ef1d024a76f21a1d40e30c302-Paper.pdf>).
> We selected this approach instead of post-hoc calibrators such as temperature scaling because incorporating focal loss allows us to address both aspects:
> (1) a non–post-hoc, uncertainty-aware, and calibration-oriented training modification,
> (2) keeping all calibration analysis fully post-hoc and consistent with the realistic constraints faced by clinical evaluators, and
> (3) the authors showed that this method is much more effective than temperature scaling.
>
> We are grateful for the strong score you provided. We have updated the manuscript by introducing these additional methods and improving the introduction to more clearly explain our emphasis on post-hoc evaluation by clinical researchers/regulators, the real-world clinical context, and the rationale for not including calibration tuning.
>
>
>
> ---
> ## comment part2:
>
> Why was the calibration set drawn from the test sets rather than from held-out training/validation data?
>
> ## Response Part2:
>
> Great observation. This was a deliberate methodological choice to reflect a realistic development workflow in which a model is trained in the “lab” and deployed/evaluated on data from a different distribution—the “clinic.” In practice, clinical evaluators only have access to the deployed model and the clinical test population, not the original training or validation data. Therefore, drawing the calibration set from the same distribution as the test set (while keeping it separate) best simulates this lab-to-clinic gap.
> We have clarified this motivation in the dataset section of the appendix.
>
>
>
> ---
> ## Comment Part3:
> It would be useful if the authors provided reliability diagrams (before/after best calibrator) with ECE bins fixed across methods.
>
> ## Response Part3:
>
> We agree that such figures would be very helpful. However, as described above, post-hoc calibration is outside the scope of our study. We only included conformal prediction because it serves both as an uncertainty quantification method and as a calibration mechanism. Its calibration effect (before/after) is demonstrated in Figure 5, which shows a substantial improvement.
>
> ---
> We hope our responses and revisions address your concerns, and we respectfully ask that you consider updating our score.
> Thank you, and please let us know if any other comments or concerns remain.

---

### Official Review · Reviewer_oGkH · 2025-11-01

**Soundness:** 3
**Presentation:** 3
**Contribution:** 3
**Rating:** 4
**Confidence:** 4

**Summary:**

The authors provide a benchmark for understanding the predicted uncertainty in a highly relevant retinal screening task including the popular diabetic retinopathy detection. The authors just compare the 6 widely used methods for uncertaintiy quantification like the Monte-carlo dropout, test-time augmentations and use the already proposed uncertainty extraction mechanisms for these methods to calculate unertainty and orgainize it as a proper benchmark to accelerate the development of reliable models in this field. Mostly they use the existing labelled datasets for popular tasks like diabetic retinopathy but also include data from some hospital for the macular and galucoma task and evaluate them under setups like domain shift by testing on local clinical dataset. Based on this, the authors draw important conclusions for the effectiveness of these methods. The setups are the usual selective prediction and calibration analysis.

**Strengths:**

Overall the analysis can be quite useful for enhancing the research in this direction, since the authors have interesting insights like Glaucoma benefits the most from this uncertainty based selective prediction or deep ensembles emerge as a reliable approach and can decently decompose uncertainty into aleatoric and epistemic components. Also findings like the uncertainty estimates become unreliable under domain shift are intersting. Section 3.5 discussing disagreement analysis among the physicians is also an important point of discussion for this setup.

**Weaknesses:**

The authors have considered very simple methods and have not considered more recent methods for selective classification like the SelectiveNet or Self-adaptive training and have also missed some other works like Deep Gamblers since it is still not exactly clear which of these approaches will work the best for these tasks and whether they are a better alternative to these considered simple method. Also there was another recent paper [1] which proposed extensions to large language models to make them more relialble which also considered the Diabetic Retinopathy dataset and showed advantages for the selective classification setup. Secondly the authors have also missed out on popular calibration methods like focal loss [2]. Like the current benchmarks seem to consider somewhat standard/older methods and is not entirely convincing how much effectively (or not) the current best models can solve this problem.

[1] Plex: Towards Reliability using Pretrained Large Model Extensions
[2] Calibrating Deep Neural Networks using Focal Loss

**Questions:**

Please see the weaknesses section. A major question is why only such limited baselines were considered which are not very recent and so it cannot directly indicate the state of model development for these tasks? Also are there any other possible tasks that can be included to make this study more comprehensive or maybe like does these tasks indicate something more about other relevant problems similar in nature?

---

> ### Author Response · Authors · 2025-11-24
> **Detailed Response**
>
> ## Comment Part1:
> Weaknesses:
> The authors have considered very simple methods and have not considered more recent methods for selective classification like SelectiveNet or Self-Adaptive Training and have also missed some other works like Deep Gamblers, ....
> ...
> ...
> ... solve this problem.
>
> [1,2]
>
> Questions:
> Please see the weaknesses section. A major question is why only such limited baselines were considered, which are not very recent, and so the results cannot directly indicate the state of model development for these tasks.
>
>
> ## Response Part1:
>
> Thank you for the thoughtful comments and for reading our manuscript carefully. We agree that many recent selective-classification and uncertainty-aware training methods exist (e.g., SelectiveNet, Deep Gamblers, Self-Adaptive Training, Plex, and focal-loss–based calibration approaches).
> However, **our goal was to highlight the importance of this topic and to provide clear guidance for clinical regulators and researchers on how to evaluate AI models in terms of risk–coverage–accuracy**.
> Therefore, our study focuses intentionally on **post-hoc evaluation**, which is essential for our clinical use case: in many real-world deployments, model testers or clinical teams do not have access to the original training data, training code, or model parameters. Methods requiring retraining, end-to-end modification, or access to internal model representations cannot be applied in these settings, which excludes approaches such as SelectiveNet, Deep Gamblers, and Plex.
>
> **That said, your comment is still valuable.** While many of the mentioned approaches are incompatible with our strict post-hoc assumption, the **focal-loss training paradigm**(ref. 2 in comment section) you suggested (especially FLSD-53, which is also uncertainty-aware) is a retraining-based calibration method that can be readily incorporated into our models. Based on your suggestion, **we have now added** experiments using FLSD-53 to examine whether uncertainty-aware, calibration-oriented training improves downstream post-hoc calibration and selective prediction performance. These new results strengthen the study and allow comparison between purely post-hoc and training-time calibration strategies.
>
> To our knowledge, this is the first large-scale benchmark using clinical ophthalmic images to characterize these trade-offs (risk–coverage–accuracy) in a realistic scenario. The closest work is the Google paper presented at NeurIPS 2021 (<https://datasets-benchmarks-proceedings.neurips.cc/paper/2021/file/ac1dd209cbcc5e5d1c6e28598e8cbbe8-Paper-round2.pdf>), which only evaluated selective prediction on diabetic retinopathy. However, their benchmark was conducted on a much smaller dataset and the discussion remained high-level—they did not analyze risk–coverage–accuracy trade-offs or systematically evaluate performance across different uncertainty thresholds. In contrast, our work presents the first large-scale, clinically grounded evaluation explicitly focused on these post-hoc trade-offs using ten times more retinal images and multiple disease categories. We also investigated domain shift in clinical settings and physician disagreement as a source of aleatoric uncertainty.
>
> ---
> ## Comment Part2
>
> Also, are there any other possible tasks that can be included to make this study more comprehensive, or do these tasks indicate something about other relevant problems of similar nature?
>
> ## Response Part2:
>
> Regarding other possible tasks: our benchmarking framework and shared scripts are designed to be extensible so that other researchers (especially clinicians) can perform further post-hoc evaluations using additional datasets or disease domains. While we could expand to other medical areas—for example, radiology or pathology—our current benchmark already includes three major retinal diseases and three deep learning model families. Importantly, ___our results show that the optimal risk–coverage–accuracy balance is disease-dependent, model-dependent, and UQ-method–dependent, meaning adding more diseases/organs/modalities would likely reinforce, rather than change, the overall conclusion.___
>
> ## Summary of our response:
> In summary, based on your suggestion, we also incorporated the focal-loss training paradigm (FLSD-53) to examine whether uncertainty-aware and calibration-oriented training can provide additional improvements. However, other uncertainty-aware approaches that are not post-hoc fall outside the scope of our study (clinical evaluators), as they require retraining or access to training data. We appreciate the suggestion and believe the updated manuscript now better clarifies the scope and rationale behind our method selection, along with the newly added FLSD-53 results.
>
> We hope that our detailed responses and the improvements made to the manuscript justify a higher score, and we respectfully ask that you consider raising our rating.
> Thank you, and please let us know if any other comments or concerns remain.

---

> > ### Comment · Reviewer_oGkH · 2025-11-26
> >
> > Thanks for a detailed response! I still believe a bit more exhaustive evaluation should have been considered and not just limiting to this post-hoc evaluation. Furthermore, the reviewers have not commented why they have refrained from using larger models including the reliability framework proposed in a recent paper [1] I pointed in the weakness section, which also analyzes various uncertainty method. I thank the reviewers for adding the FSLD experiments but based on the justifications provided which only partially addresses my concerns, I would like to maintain my original ratings.
> >
> > [1] Plex: Towards Reliability using Pretrained Large Model Extensions

---

### Author Response · Authors · 2025-11-24
**General Response**

We thank the reviewers for their thoughtful feedback, which helped us strengthen the manuscript substantially. In response to concerns about limited baselines and recent methods, we clarified that our study intentionally focuses on **post-hoc evaluation**, reflecting the realistic **clinical setting where evaluators** do not have access to training data or the ability to retrain vendor-provided models. Our revisions also emphasize our objective, which is the importance of **risk–coverage–accuracy analysis** and the associated trade-offs.

Nevertheless, based on reviewer suggestions, we **expanded our experiments** to include the focal-loss–based FLSD-53 model, updated all figures, and added new analyses to compare post-hoc versus calibration-aware training. We also clarified why retraining-dependent approaches (e.g., SelectiveNet, Deep Gamblers, Plex, Laplace approximation, SWAG, evidential learning) fall outside our strict clinical-evaluation scope. In addition, we extended the manuscript with detailed rationale for hyperparameter choices, **expanded our external clinical dataset** from 536 to ~743 images, clarified calibration-set selection, and included new explanations regarding domain-shift causes and potential adaptation strategies. We also **improved the Introduction and Scope sections** to clearly articulate our benchmarking goals, expanded our justification for using ViT-based architecture families given ViT-based retina foundation models, and updated all related discussions accordingly.

We believe these clarifications and additions address the reviewers’ concerns and significantly enhance the rigor, transparency, and usefulness of the work.
The **revised manuscript is also submitted**, with all added or adjusted sections **highlighted in blue**.


**References mentioened in responses**

[1] N. Band, et al. Benchmarking bayesian deep learning on diabetic retinopathy detection tasks. Neurips 2021: arXiv
preprint arXiv:2211.12717, 2022.

[2] Y. Gal et al. Dropout as a bayesian approximation: Representing model uncertainty in deep learning. ICML 2016

[3] T. Itoh, et al. Multi-task deep learning for predicting
metabolic syndrome from retinal fundus images in a japanese health checkup dataset. Plos one

[4] B. Lakshminarayanan, et al. Simple and scalable predictive uncertainty estimation using deep ensembles. NIPS’2017

[5] Y. Li, et al. Automated detection of myopic maculopathy in mmac 2023: achievements in classification, segmentation, and spherical equivalent prediction. MICCAI 2023.

[6] D. Milanés-Hermosilla, et al. Monte carlo dropout for uncertainty estimation and motor imagery classification. Sensors, 2021.

[7] N. Moshkov, et al. Test-time augmentation
for deep learning-based cell segmentation on microscopy images. Scientific reports, 2020.

[8] W. Nazzal,  et al. Improving medical image segmentation
using test-time augmentation with medsam. Mathematics, 2024.

[9] Y. Ovadia, et al. Can you trust your model’s uncertainty? evaluating predictive uncertainty under
dataset shift. Advances in neural information processing systems, 2019.

[10] Y. Sun, et al. Test-time training with self-
supervision for generalization under distribution shifts. ICML 2020.

[11] D. Wang, et al. Tent: Fully test-time adaptation
by entropy minimization. 2020.

[12] K.-L. et al. Deep learning with test-time augmentation for radial endobronchial
ultrasound image differentiation: a multicentre verification study. BMJ, 2023.

[13] K. Zhou, et al. Domain generalization: A survey. IEEE
transactions on pattern analysis and machine intelligence,  2022.

---

### Meta-Review · Area_Chair_2BUH · 2026-01-06

**Summary:**

This paper presents a benchmark for uncertainty quantification for retinal disease screening. While the authors seem to have some contribution in experiments, the novelty of the work is limited. The reviewers have diverse scores from 2 to 8 to this paper. I am more toward the rejection side as the technical contribution is very limited in this paper. On the other side, clinical validation of AI retinal disease screening does not match with ICLR very well. Therefore, I suggest the authors to submit this work to other venue.

**Reviewer Concerns:**

The reviewers have made good evaluation of the work that it is a scientifically sound evaluation of the status quo and a clear, end-to-end benchmark. However, this is not the main purpose for ICLR, which focuses on novel contribution in methodology.

**Reviewer Scores:**

The work received diverse scores. But I believe the decision to this paper is not depending on the scores but the contribution and relevant to this conferences.

---

### Decision · Program_Chairs · 2026-01-26

Reject